# Reshaping Reservoirs: Hebbian Plasticity for Improved Data Separability

## Abstract

This paper introduces Hebbian Architecture Generation (HAG), a method grounded in Hebbian plasticity principles, designed to optimize the structure of Reservoir Computing networks. HAG adapts the synaptic weights in Recurrent Neural Networks by dynamically forming connections between neurons that exhibit high Pearson correlation. Unlike conventional reservoir computing models that rely on static, randomly initialized connectivity matrices, HAG tailors the reservoir architecture to specific tasks by autonomously optimizing network properties such as signal decorrelation and singular value spread. This task-specific adaptability enhances the linear separability of input data, as supported by Cover's theorem, which posits that increasing the dimensionality of the feature space improves pattern recognition. Experimental results show that HAG outperforms traditional Echo State Networks across various predictive modeling and pattern recognition benchmarks. By aligning with biological principles of structural plasticity, HAG addresses limitations of static reservoir architectures, offering a biologically plausible and highly adaptable alternative for improved performance in dynamic learning environments.

## 1 Introduction

In this paper, we introduce Hebbian Architecture Generation (HAG), a novel approach that dynamically adjusts the synaptic weights in Recurrent Neural Networks (RNNs) to improve the quality of their representations. HAG is founded on Hebbian theory (Attneave et al., 1950), the principle that synaptic connections between co-activating neurons strengthen over time, encapsulated by the maxim "neurons that fire together wire together." Leveraging this idea of topology informed by correlations, HAG dynamically constructs reservoir connectivity to better suit the task at hand, addressing several core challenges in Reservoir Computing (RC).

RC, particularly through Echo State Networks (ESNs) (Jaeger, 2001), provides a framework for transforming input signals into high-dimensional dynamic states, as dictated by Cover's theorem (Cover, 1965). Cover's theorem posits that increasing the dimensionality of a feature space enhances the likelihood that complex data patterns become linearly separable, which is crucial for efficient learning in RC. However, traditional ESNs, as highlighted by Jaeger (2005), face notable limitations in reservoir suitability, unsupervised adaptation, or alignment with biological principles.

HAG addresses these limitations in several ways:

1. **Task-specific:** beginning with a blank connectivity matrix, HAG forms dynamic synaptic connections, which inherently tailors the reservoir to the demands of specific tasks, moving beyond the random, static connectivity found in traditional ESNs.

2. **Unsupervised adaptation:** It autonomously optimizes network properties such as singular value spread and signal decorrelation by leveraging high Pearson correlations between neurons, thereby enhancing the linearity of the feature space.

3. **Performance predictors:** By focusing on measurable properties such as signal decorrelation and feature space expansion, we provide specific criteria for assessing and enhancing reservoir performance. This approach moves beyond the tautological notion that a reservoir is suitable if it yields accurate models.

4. **Biological insights:** Reflecting the principles of Hebbian and structural plasticity, HAG mimics the adaptability of biological neural networks by forming new connections and reorganizing correlated input features to optimize feature extraction.

Our experimental results demonstrate that HAG not only addresses those key challenges, but also improves performance across multiple benchmarks compared to traditional RC approaches.

This paper first presents the necessary background (section 2) for echo state networks. We then introduce the HAG algorithm, the motivation behind this rule, and the main elements of our experiments in Section 3. Section 4 then covers the significant performance improvements on several benchmarks for two versions of the HAG algorithm compared to traditional RC. Finally, in section 5 we show that our algorithm projects input data into a higher-dimensional space which leads to better performance according to Cover's theorem.

## 2 BACKGROUND

### 2.1 ECHO STATE NETWORKS

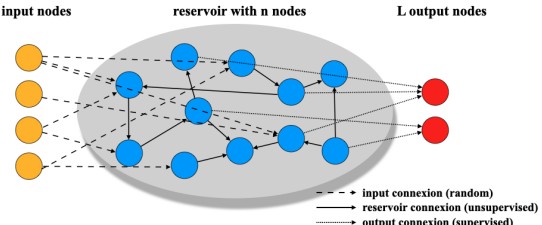

Figure 1: Schematic architecture of ESNs

In ESNs, the challenge of capturing nonlinear relationships in data is addressed by transforming the time series input into a high-dimensional space using a dynamical system (the reservoir) which in the case of ESNs is an artificial neural network of recurrently connected nodes. In this new feature space, the complex, nonlinear relationships may become linear, enabling the use of simple linear models, such as ridge regression, to accurately estimate the target variable. Notably, only the linear readout layer is trained, while the reservoir remains fixed, reducing computational complexity and enhancing training efficiency.

An illustration of the ESN framework (Lukoševičius & Jaeger, 2009) (Jaeger, 2001) is given in Fig. 1. It normally consists of a randomly initiated RNN of $n$ neurons (the *reservoir*), and a trained *readout layer* that creates the outputs as a linear combination of the reservoir neuron states.

The evolution of the vector states $\mathbf{x}$ of the $n$ neurons is determined by the interactions between the connection matrix $\mathbf{W}$, the input matrix $\mathbf{W_{in}}$, the input $u[t]$ and the activation function $\sigma$ that we fix as the hyperbolic tangent function, combined in the following equation:

$$\mathbf{x}[t+1] = \sigma(\mathbf{W} \times \mathbf{x}[t] + \mathbf{W_{in}} \times u[t] + \mathbf{b}) \tag{1}$$

The purpose of the bias term $\mathbf{b}$ is to enrich the dynamics of the network, but is kept small through bias scaling to avoid dominating the system. $\mathbf{b}$ is a $1 \times n$ vector and dimension of the matrices are $n \times n$ for $\mathbf{W}$ and $1 \times n$ for $\mathbf{W_{in}}$. In ESNs, these weights are typically randomly initialized from a chosen distribution, and modulated by input scaling ($s_{in}$) and bias scaling ($s_b$) that are two hyperparameters of the systems. However, as we explain below, mean-HAG uses a different weight initialization scheme where these weights evolve during operation according to an unsupervised rule.

Once $\mathbf{W}$, $\mathbf{W_{in}}$ and $\mathbf{b}$ are fixed, the desired output is obtained through the following equation :

$$y[t+1] = \mathbf{W_{out}} \times \mathbf{x}[t+1] + \mathbf{b_{out}} \tag{2}$$

where $\mathbf{W_{out}}, \mathbf{b_{out}}$ are learned using ridge regression with regularization parameter $\lambda$ to prevent overfitting by controlling the size of the output such that the generated train output, $y[t]$, optimally approximates a desired target output, $y^{target}[t]$ (Lukoševičius & Jaeger, 2009).

## 3 METHODS

While Lukoševičius & Jaeger (2009) gives an overview of other unsupervised methods used to improve RC, a detailed overview of the effect of biological rules on the dynamics of RC is given in Morales et al. (2021). Additionally, the self-organizing recurrent network (SORN) introduced by Lazar (2009) focuses on spiking neurons and leverages multiple forms of plasticity to adapt a form of spiking neural network, Cazalets & Dambre (2023) on structural plasticity in reservoirs with limited results and Schrauwen et al. (2008) use synaptic plasticity in an unsupervised manner.

Our approach differs from previously explored techniques such as Cazalets & Dambre (2023) or Schrauwen et al. (2008), on two points. First, our algorithm takes inspiration from structural plasticity and starts from a blank connectivity matrix, and is then able to create connections between neurons that are not connected. Second, our algorithm is based on the Pearson correlation, which allows us to recombine frequently correlated input features, transforming the data into a new feature space with reduced correlation. In this transformed space, the input features are recombined into a higher-dimensional space where patterns become as linearly independent as possible.

### 3.1 HAG ALGORITHM

We start from a blank echo state network with no connections except from the input connectivity. The HAG algorithm dynamically adjusts synaptic weights $w_{ij}$ based on either variance or average of the activity of the reservoir's neurons. Those two types of homeostatic mechanisms lead to two corresponding algorithms:

1. **Mean Homeostasis Function (mean-HAG)**: Adjusts synaptic weights based on deviations from a target mean activity rate.

2. **Variance Homeostasis Function (variance-HAG)**: Adjusts synaptic weights by comparing the standard deviation of neuron states to a target standard deviation.

A more detailed explanation on the HAG algorithm is provided in Appendix B with pseudo-code in B.1.

Every $T$ time steps, we calculate for each neuron a growth indicator:

$$\Delta z_i = \frac{1}{\beta}(s_i - \rho) \tag{3}$$

where $s_i$ represents either the average $i$-th neuron's activity $\langle x_i \rangle_T$ over period T for mean-HAG, or the $i$-th neuron's standard deviation $\sigma_{x_i,T}$ over period T for variance-HAG. We denote $\rho_r$ as the activity target value and $\beta_r$ as the rate spread (for mean-HAG), and $\rho_v$ as the variance target and $\beta_v$ as the variance spread (for variance-HAG).

If $\Delta z_i < -1$, the neuron needs to increase its activity. In this case, one connection weight is increased by $\delta w$. The creation of new connections is restricted to neurons that have been identified as requiring additional connections. To choose which connection to increase, for every neuron that has not yet achieved homeostasis, we compute pairwise Pearson correlation coefficients (Pearson, 1895) with every other neuron that is also not at homeostasis. We establish a connection with the highest correlated neuron. Detailed description can be found in Appendix B.2.

If $\Delta z_i > 1$, the neuron needs to decrease its activity. In this case, one connection weight is decreased by $\delta w$. Unlike the creation of new connections, the pruning of connections is performed randomly, independently of the state of the neuron's partners, regardless of whether they also need to decrease their activity or not.

The network is said to be at homeostasis if, for each neuron $i$, the absolute value of $\Delta z_i$ is less than 1 (i.e. $s_i$ is between $\rho - \beta$ and $\rho + \beta$). At this homeostasis the network maintains a desired level of variance or average in neuronal activity as seen in Figure 2a.

We impose a limit on the degree of the node, $\gamma$. By design, the RNN will only have positive connections, which ensures predictable increases in neuronal activity. This configuration restricts the network to excitatory connections, which might limit the computational properties of the ESN.

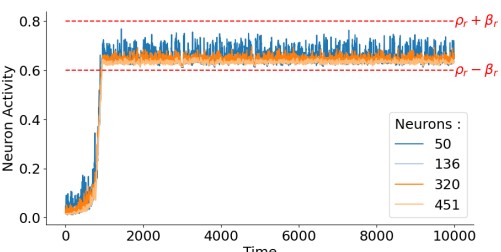
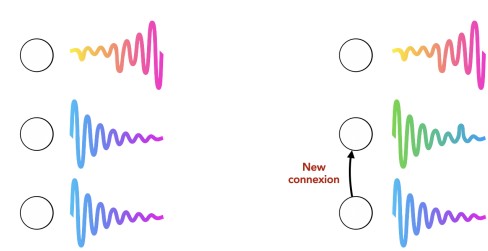

(a) Different neurons' activities during training with the mean-HAG algorithm. As new connections are added every $T$ period, the average neuron states $\langle x_i \rangle_T$ clearly converge within the homeostatic range of activity.

(b) Nodes are initially correlated (on the left), creating a connection between them reduce redundancy and produce decorrelated, more informative states for improved reservoir representation (on the right).

Figure 2: (a) Neuronal activities during training with the mean-HAG algorithm. (b) Illustration of how connecting correlated nodes improves reservoir efficiency.

### 3.1.1 ENSURING CONVERGENCE

The mean-HAG algorithm dynamically adjusts neural connectivity based on deviations from target mean activity rates. This ensures convergence during the learning phases as seen in Figure 2a.

In contrast, the variance-HAG algorithm aims to increase variability, which could inadvertently amplify the overall signal strength and lead to potential saturation. To counteract this, we employ a homeostatic plasticity mechanism, when a neuron's activity level exceeds a saturation threshold $\theta_{\text{sat}}$, the algorithm scales the weights by $\eta_{\text{sat}}$. This adjustment is crucial for maintaining network balance, avoiding disruptive saturation, and ensuring the smooth convergence of the algorithm.

### 3.1.2 MOTIVATION

Reservoir Computing (RC) leverages the concept of transforming input data into a higher-dimensional space where complex patterns become more linearly separable. This idea is rooted in Cover's theorem, which suggests that projecting data into a high-dimensional nonlinear space increases the likelihood of linear separability. Traditional Echo State Networks (ESNs) accomplish this transformation using randomly instantiated reservoirs, a method described by Lukoševičius (2012) as "the antithesis of the 'optimal.'". Such random reservoirs fail to exploit the structure of the input data, potentially limiting their effectiveness in separating complex patterns.

Our Hebbian Architecture Generation (HAG) algorithm is motivated by the need to enhance this transformation process. By dynamically adjusting the reservoir's connectivity based on neuron activations, HAG aims to design a network structure that recombines the inputs to effectively increases the dimensionality and richness of the feature space, a mechanism illustrated by Figure 2b.

Drawing inspiration from biological systems, we incorporate principles of structural plasticity observed in neural circuits, where neuronal activity leads to both growth and retraction of synaptic connections (Fauth & Tetzlaff, 2016; Cohan & Kater, 1986; Vaillant et al., 2002). Such plasticity mechanisms dynamically shape neural networks in response to stimuli, enhancing their adaptability. Similar mechanisms have been shown to drive transitions between dynamic states in biological networks due to changes in input strength, mediated by homeostatic plasticity (Zierenberg et al., 2018). By integrating unsupervised, activity-dependent structural plasticity with minimal supervised elements, our approach enhances the efficiency and effectiveness of reservoir computing systems. This aligns with the hypothesis proposed by Zador (2019), suggesting that biologically inspired adaptations can achieve remarkable efficiency and performance with significantly less supervision compared to conventional neural network models. These findings underscore the importance of adaptive connectivity in optimizing dynamic representations.

## 3.2 DATASETS

ESNs have employed a diversity of benchmarks and datasets, as extensively documented in Sun et al. (2020). Our study employs a diverse array of datasets to assess the capabilities of our algorithm.

We utilized `ReservoirPy` (Trouvain et al., 2020), a library updated with contemporary advancements, featuring a modular architecture for assembling ESNs and a suite of standard algorithms for training the readout layer.

### 3.2.1 TASKS

The training of the ESN system occurs in two phases. Initially, for classification tasks such as speech recognition, the reservoir processes input signals into high-dimensional representations, capturing the dynamic state at the end of each input sequence. This state, representing the reservoir's response to the input, serves as the feature vector for training the readout layer using ridge regression to determine the optimal output weights $W_{\text{out}}$.

For prediction tasks aimed at forecasting future values from past inputs, we adopt a sequence-to-sequence approach. The reservoir updates its state at each time step, using the entire sequence of states to train the readout layer. Here, $W_{\text{out}}$ is optimized to minimize the difference between the predicted outputs and the actual targets at each time step.

### 3.2.2 CLASSIFICATION DATASETS

**CatsDogs**   The **CatsDogs** dataset is the auditory counterpart to the classic image classification task, containing WAV audio files—164 for cats (1,323 seconds) and 113 for dogs (598 seconds)—recorded at 16 kHz.

**SpokenArabicDigits**   The **Spoken Arabic Digits** dataset contains recordings of 88 individuals pronouncing Arabic digits 0–9, with ten pronunciations per digit per speaker. It is commonly used for testing speech recognition algorithms due to the phonetic diversity of Arabic numerals. (Mouldi Bedda, 2008)

**Japanese Vowels**   The **Japanese Vowels** dataset includes recordings of nine male speakers pronouncing sequences of Japanese vowels. It is frequently utilized in research on linguistic characteristics and speaker identification technologies. (Mineichi Kudo, 1999)

**FSDD**   The **Free Spoken Digit Dataset** (FSDD) is an open collection of English audio recordings of spoken digits 0–9 by multiple speakers. Designed for experimenting with speech processing techniques like classification and clustering, it provides a straightforward entry point into digital speech processing. (Jackson et al., 2018)

**SPEECHCOMMANDS**   The **SPEECHCOMMANDS** dataset comprises over 105,000 audio files of short commands like "Yes," "No," "Up," and "Down," spoken by various speakers. Widely used for training and benchmarking models in voice user interfaces. (Warden, 2018)

### 3.2.3 PREDICTION DATASETS

**MackeyGlass**   Derived from a differential equation, the **MackeyGlass** dataset is noted for its use in modeling nonlinear dynamics and chaos, making it a challenging dataset for time series prediction models. (Mackey & Glass, 1977)

**Lorenz**   The **Lorenz** dataset is based on the Lorenz attractor, a set of chaotic differential equations used extensively in predicting nonlinear system behaviors and atmospheric studies. (Lorenz, 1963)

**Sunspot (SILSO)**   The **Sunspot** dataset from SILSO includes smoothed monthly mean sunspot numbers from 1749 to 2020, reflecting solar activity and serving as a proxy for the Sun's magnetic field strength. Its complexity makes it a significant test case for forecasting models in time series analysis and solar studies. (SILSO World Data Center, 2024)

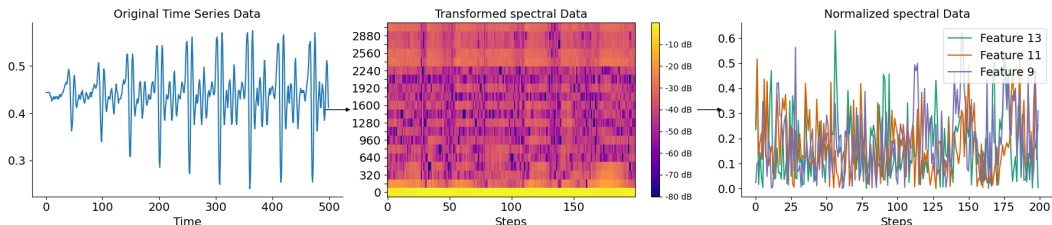

Figure 3: Preprocessing pipeline for classification datasets (here for FSDD) from raw time series (left) to spectrogram (middle) and normalized features (right).

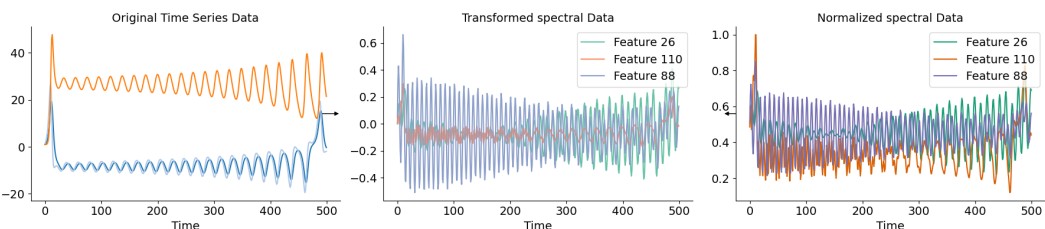

Figure 4: Preprocessing pipeline for prediction datasets (here for Lorenz) from raw time series (left) to features after filtering around main frequencies (middle) and normalized features (right). Here the number of time steps in the time series are preserved to perform prediction tasks

### 3.3 PREPROCESSING

We adopt distinct preprocessing methods tailored to classification and prediction tasks, each leveraging the properties of audio signals to prepare data for analysis and model training.

In the classification tasks, we preprocess the audio signals by converting them into a spectral representations using *Mel-Frequency Cepstral Coefficients* (MFCCs), computed with the `librosa` library (McFee et al., 2015). MFCCs are widely used in speech and audio processing to capture the timbral aspects of sound by modeling the human auditory system's response (Davis & Mermelstein, 1980; Verstraeten et al., 2005). They provide a compact representation of the spectral properties of audio signals, making them suitable for classification tasks. This process is illustrated in Figure 3.

Conversely, for prediction tasks, the preprocessing utilizes band-pass filters to isolate specific frequency bands from the signal. This approach starts with the identification of peak frequencies. Frequency bands are then defined around these peaks by calculating the bandwidth as half the distance to adjacent peak frequencies. This method effectively segments the time series into parts that contain relevant information for predictive modeling. The filtered signals obtained from this process constitute the feature set used for forecasting future events or states from past time series. Compared to the method for classification tasks, this method preserves the number of time steps in the time series, which is essential to perform prediction tasks. This process is illustrated in Figure 4.

For each dataset, the network size is set to just above 500 units by duplicating the multivariate input a specific number of times. For example, the SpokenArabicDigits dataset uses a network size of 507, achieved by duplicating its 13-dimensional spectrogram 39 times. Similarly, the Japanese Vowels dataset has a network size of 504, with a 12-dimensional spectrogram duplicated 42 times.

### 3.4 HYPERPARAMETER SEARCH

To ensure robust model evaluation, we employed cross-validation strategies appropriate for each task type. For classification tasks without predefined groups, we used Stratified K-Fold cross-validation with shuffling to maintain class distribution across folds. When group-based classification was necessary, Stratified Group K-Fold cross-validation was applied to preserve both class distribution and group integrity, preventing data leakage. For time series prediction tasks, Time Series Split cross-validation was utilized to respect temporal ordering and prevent future data leakage.

Table 1: Mean cross-validation classification accuracy averaged over 3-folds for 400 trials.

|                    | E-ESN     | ESN     | IP-ESN    | mean-HAG  | variance-HAG |
|--------------------|-----------|---------|-----------|-----------|--------------|
| CatsDogs           | 61.6%     | 61.5%   | **83.0%** | 71.4%     | 76.8% |
| Japanese Vowels    | 95.2%     | 95.9%   | **98.5%** | 96.3% | 94.4%     |
| SpokenArabicDigits | 56.1%     | 76.6%   | 71.4%     | 77.0% | **82.8%** |
| FSDD               | 28.0%     | 20.0%   | **72.9%** | 51.4% | 44.6%     |
| SPEECHCOMMANDS     | 5.8%      | 5.91%   | 7.1%      | 32.6% | **34.0%** |

Table 2: Cross-validation NRMSE averaged over 3-folds for 400 trials

|             | E-ESN      | ESN      | IP-ESN    | mean-HAG    | variance-HAG |
|-------------|------------|----------|-----------|-------------|--------------|
| MackeyGlass | 0.070      | 0.068    | **0.055** | 0.066       | 0.063 |
| Lorenz      | 0.20       | 0.14 | 0.147  | 0.21        | **0.14**     |
| Sunspot     | **0.0155** | 0.0202   | 0.0223    | 0.0195 | 0.0198     |

Hyperparameter tuning was performed using `optuna` (Akiba et al., 2019), leveraging the Tree-structured Parzen Estimator (TPE) sampler over 400 trials per dataset and algorithm variant. The TPE sampler efficiently explores the hyperparameter space by focusing on promising regions, making it suitable for our optimization tasks (Bergstra et al., 2011).

The hyperparameters optimized included input scaling, bias scaling, ridge regression coefficient, and algorithm-specific parameters such as rate target and variance target. Table 5 summarizes the hyperparameter ranges.

## 4 RESULTS

We evaluated the performance of the HAG algorithms (mean-HAG and variance-HAG) against baseline models, including traditional ESNs, ESNs with only positive weights (E-ESNs), and Intrinsic Plasticity ESNs (IP-ESNs) as described in Schrauwen et al. (2008). The results highlight the robust improvements offered by the HAG algorithms, particularly in classification tasks. Details on the hyperparameter search results can be found in C.2.

### 4.1 CROSS VALIDATION PERFORMANCES

Tables 1 and 2 present the mean cross-validation classification accuracies and normalized root mean square errors (NRMSEs), respectively, averaged over 3-folds for 400 trials per model and dataset.

In classification tasks, the HAG algorithms consistently outperformed E-ESN and standard ESNs except for instance of the JapaneseVowels eplained by an ill-conditioning of the reservoir (see Appendix B.3). For example, on the SPEECHCOMMANDS dataset, variance-HAG achieved a notable accuracy of 34.0%, significantly outperforming the ESN (5.9%) and E-ESN (5.8%). The IP-ESN model demonstrated strong performance on the CatsDogs, Japanese Vowels datasets and FSDD, achieving the highest accuracies of 83.0%, 98.5% and 72.9%, respectively. However, on datasets such as SpokenArabicDigits and SPEECHCOMMANDS, HAG algorithms surpassed IP-ESN, showing greater adaptability to diverse datasets.

In prediction tasks, variance-HAG achieved competitive performance, matching the standard ESN on the Lorenz dataset (NRMSE of 0.140). However, the IP-ESN model achieved the lowest NRMSE on the MackeyGlass dataset (0.055), highlighting its strength in this type of task. On the Sunspot dataset, E-ESN achieved the best NRMSE of 0.0155, but both HAG algorithms performed comparably, demonstrating their versatility across different prediction challenges.

Table 3: Test classification accuracy over 8 trials on the test dataset using the whole training dataset

| | E-ESN | ESN | IP-ESN | mean-HAG | variance-HAG |
|---|---|---|---|---|---|
| CatsDogs | 60.9% | 60.4% | 55.41% | 68.8% | **69.1%** |
| Japanese Vowels | 95.8% | 96.7% | 97.0% | **98.4%** | 94.19% |
| SpokenArabicDigits | 63.6% | 74.8% | 57.84% | **95.4%** | 95.2% |
| FSDD | 23.5% | 25.3% | 32.75% | **46.3%** | 44.0% |
| SPEECHCOMMANDS | 6.11% | 6.94% | 6.35% | 17.3% | **27.4%** |

Table 4: Mean NRMSE over 8 trials on the test dataset using the whole training dataset

| | E-ESN | ESN | IP-ESN | mean-HAG | variance-HAG |
|---|---|---|---|---|---|
| MackeyGlass | **0.0518** | 0.0533 | 0.0734 | 0.0667 | 0.0663 |
| Lorenz | 0.174 | 0.187 | 0.174 | 0.173 | **0.159** |
| Sunspot | 0.0695 | 0.0376 | 0.0510 | **0.0342** | 0.0544 |

## 4.2 TEST PERFORMANCES

Tables 3 and 4 summarize the models' test performance using the best hyperparameters identified during cross-validation.

On the test datasets, the HAG algorithms maintained strong performance. On the SpokenArabicDigits dataset, mean-HAG achieved a remarkable accuracy of 95.4%, significantly outperforming IP-ESN (57.8%). On the SPEECHCOMMANDS dataset, variance-HAG achieved the highest accuracy of 27.4%, a substantial improvement over all baselines. IP-ESN excelled on the CatsDogs, Japanese Vowels and FSDD dataset during cross-validation but underperformed on the test set compared to the HAG algorithms, suggesting potential overfitting.

In prediction tasks, variance-HAG achieved the lowest NRMSE on the Lorenz dataset (0.159), while mean-HAG performed best on the Sunspot dataset (0.0342). IP-ESN underperformed on the test datasets.

The results demonstrate that the HAG algorithms, particularly variance-HAG, consistently outperform baseline models, including IP-ESN, in classification tasks. Their ability to dynamically adjust synaptic weights based on Hebbian principles allows for improved representation of input data, enhancing accuracy and generalization. While IP-ESN achieved strong performance on some datasets, its inconsistency across tasks highlights the advantage of HAG's tailored reservoir dynamics, particularly on the most complex task of SPEECHCOMMANDS. In prediction tasks, HAG algorithms showed competitive performance but did not consistently outperform E-ESN or IP-ESN. This suggests that further optimization of the HAG approach is needed for time-series forecasting applications.

## 5 DISCUSSION

We hypothesize that our algorithm enhances the dynamics of ESNs by transforming redundant inputs into more informative representations. Leveraging Hebbian learning principles, our approach projects input data into a new feature space with reduced feature correlation, enriching the pool of features and increasing the likelihood of discovering linearly separable representations. This transformation is expected to improve the network's ability to perform complex tasks by creating a richer set of features for downstream processing.

Aligned with our hypothesis based on Cover's theorem, that increasing the dimensionality of neural states simplifies problem-solving, we evaluate the dynamic richness of our reservoir (Gallicchio & Micheli, 2022), which is specifically designed to increase neural state dimensionality. We assess dynamic richness using Pearson correlation and Cumulative Explained Variance (CEV). Detailed

analysis of Pearson correlations and CEVs values for every dataset/function combination are presented in Appendix D

## 5.1 PEARSON CORRELATION

To elucidate the operational dynamics of our reservoir, we assess the correlation among neural states using Pearson correlation (Pearson, 1895), which measures the linear relationship between neuron activation states. This metric helps understand inter-neuronal connectivity and synchrony, directly impacting the network's ability to process complex data patterns. By computing Pearson correlation coefficients, we quantify initial levels of synchrony and track their evolution over time, offering insights into the network's dynamic restructuring in response to varying inputs. Detailed explanation about this calculation can be found in Appendix A.2

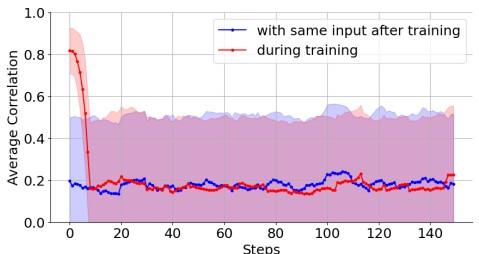
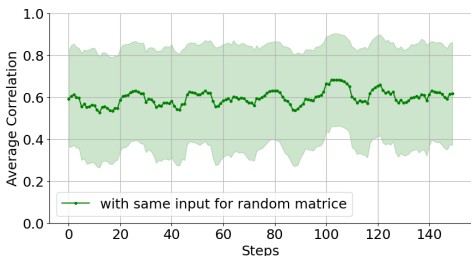

(a) during the formation of the connections for a set of good performing hyperparameters

(b) for a random reservoir instantiated with best performing hyperparameters

Figure 5: Comparison of the evolution of the average Pearson correlation between neural states between the mean-HAG-designed and randomly instantiated reservoirs on the same input data (FSDD dataset). Lower Pearson correlation indicates better conditioning of the learning problem.

Figure 5 illustrates the correlation evolution within a high-performing HAG network compared to a randomly initialized reservoir. Initially, the correlation metric is near unity, indicating highly synchronized neuron states, a typical characteristic of freshly initialized networks where states are scaled input vectors with a bias term, resulting in almost identical states. Over time, a pronounced decline in mean correlation is evident, aligning with the HAG method's objective to refine the connectivity matrix by dynamically linking neurons based on correlation levels. This fosters a more diverse and functionally rich neural dynamic, enhancing the reservoir's computational capabilities. The comparison supports our hypothesis that HAG networks better represent complex input data than traditional ESNs.

It should be noted that similar patterns are observed across different graphs, highlighting the underlying coherence in input data characteristics. The same pattern of Pearson correlation can be observed in the inputs, that then influence neural state variations.

## 5.2 CUMULATIVE EXPLAINED VARIANCE

To quantify the reservoir's dynamic behavior, we perform Principal Component Analysis (PCA) on the reservoir states. This approach allows us to analyze how the variance in the reservoir's state space is distributed among different principal components, providing insight into the complexity and richness of the dynamics.

We denote $\mathbf{H}$ as the data matrix formed by collecting the reservoir states, where each column corresponds to a state vector at a particular time step. Performing PCA on $\mathbf{H}$ yields singular values $\sigma_1, \sigma_2, \ldots, \sigma_n$, arranged in decreasing order. These singular values are directly related to the variance explained by each principal component.

The proportion of variance explained by the $j$-th principal component is calculated as:

$$R_j = \frac{\sigma_j^2}{\sum_{k=1}^{n} \sigma_k^2} \tag{4}$$

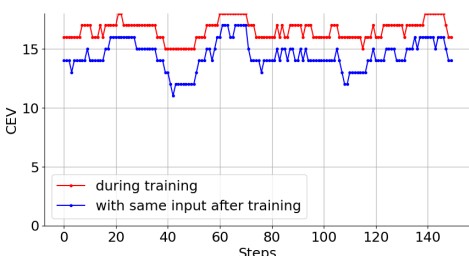 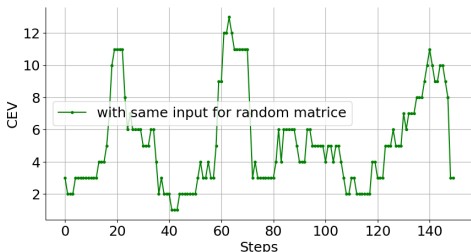

(a) during the formation of the connections for a set of good performing hyperparameters

(b) for a random reservoir instantiated with best performing hyperparameters

Figure 6: Comparison of evolution of moving cumulative explained variance (with $\theta = 0.9$) on the same input data (here with the SPEECHCOMMANDS datasets). Higher CEV indicates better conditioning of the learning problem.

The cumulative explained variance up to the $d$-th principal component is then given by:

$$C_d = \sum_{j=1}^{d} R_j = \frac{\sum_{j=1}^{d} \sigma_j^2}{\sum_{k=1}^{n} \sigma_k^2} \tag{5}$$

This cumulative measure indicates the total proportion of variance captured by the first $d$ principal components. To assess the effective dimensionality of the reservoir's state space, we determine the minimum number of principal components required to reach a predetermined threshold $\theta$ of cumulative explained variance:

$$D = \arg\min_{d} (C_d \geq \theta) \tag{6}$$

A higher value of $D$ suggests that more principal components are needed to capture the same amount of variance, indicating a richer and more complex dynamic structure within the reservoir.

Figure 6 presents the cumulative explained variance curves for both the HAG-designed reservoir and the randomly instantiated reservoir on SPEECHCOMMANDS dataset. The HAG-designed reservoir exhibits a more gradual increase in cumulative explained variance, requiring more principal components to reach the threshold $\theta$. This implies a higher effective dimensionality compared to the random reservoir, supporting our hypothesis that the HAG algorithm enhances the richness of the reservoir's dynamics.

## 6 CONCLUSION

We presented a new algorithm generating a connectivity matrix by identifying and connecting highly correlated neural nodes within a reservoir. This algorithm contrasts with traditional random matrix instantiation by creating a connectivity pattern that enhances decorrelation of reservoir states. By aligning the connectivity within the reservoir to the intrinsic correlations of the system, the proposed approach not only supports a new theoretical parallel with Hebbian plasticity but also demonstrates practical superiority over conventional methods.

The dynamic adaptability of HAG not only addresses the limitations of static reservoirs in Echo State Networks but also showcases the practical application of biologically-inspired algorithms in improving computational efficiency and task-specific performance. Our results underscore the utility of biologically-inspired design principles in computational models, emphasizing the potential of Hebbian learning rules not just as a theoretical construct but as a practical tool in machine learning.

Future research should delve deeper into understanding the mechanisms that enable HAG to perform well with complex datasets. Additionally, exploring the scalability of HAG in larger and more intricate systems will further validate its effectiveness and contribute to the development of more intelligent and adaptable neural network models.

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

# A  PEARSON CORRELATION

To dynamically form connections in the reservoir, we leverage the Pearson correlation coefficient (Pearson, 1895) to identify neurons that exhibit strong linear relationships in their activation patterns. This process is performed exclusively on neurons that have not yet achieved homeostasis, as defined by the growth indicator $\Delta z$.

## A.1  PEARSON CORRELATION COEFFICIENT

For two neurons $i$ and $j$, the Pearson correlation coefficient, $r_{ij}$, measures the linear relationship between their respective activation states, $x_i[t]$ and $x_j[t]$, over a time period $T$. It is defined as:

$$r_{ij} = \frac{\sum_{t=1}^{T}(x_i[t] - \bar{x}_i)(x_j[t] - \bar{x}_j)}{\sqrt{\sum_{t=1}^{T}(x_i[t] - \bar{x}_i)^2}\sqrt{\sum_{t=1}^{T}(x_j[t] - \bar{x}_j)^2}}, \tag{7}$$

where:

- $x_i[t]$ and $x_j[t]$ are the activation states of neurons $i$ and $j$ at time $t$,

- $\bar{x}_i = \frac{1}{T}\sum_{t=1}^{T} x_i[t]$ is the mean activation state of neuron $i$ over the period $T$,

- $\bar{x}_j = \frac{1}{T}\sum_{t=1}^{T} x_j[t]$ is the mean activation state of neuron $j$ over the same period.

The coefficient $r_{ij}$ ranges from $-1$ (perfect negative correlation) to 1 (perfect positive correlation), with 0 indicating no linear relationship.

## A.2  PEARSON AS A MEASURE OF RICHNESS

Figure 5 illustrates the average Pearson correlation between the activations of all neurons in the reservoir within each time window. Using a sliding window approach, the reservoir states are segmented into overlapping windows of a fixed size. For each window, the pairwise Pearson correlation coefficients, $r_{ij}$, are calculated for all neuron pairs $(i, j)$, excluding self-correlations $(i = j)$.

The average Pearson correlation for a given time window is defined as:

$$\mu_{\text{window}} = \frac{1}{N(N-1)} \sum_{i=1}^{N} \sum_{j=i+1}^{N} r_{ij}, \tag{8}$$

where:

- $N$ is the number of neurons in the reservoir,

- $r_{ij}$ is the Pearson correlation coefficient between neurons $i$ and $j$ over the time window.

This metric, $\mu_{\text{window}}$, represents the mean correlation between neuron activations in the window. By plotting $\mu_{\text{window}}$ for successive time windows, Figure 5 shows how the average correlation evolves during the pretraining phase.

Lower average correlations over time indicate reduced synchronization among neurons, reflecting the reservoir's increasing ability to generate a diverse and decorrelated feature space. This is a desirable property in reservoirs, as it enhances their capacity to separate input patterns in a high-dimensional space, aligning with the goals of effective reservoir design.

# B DETAILS ON THE HAG ALGORITHM

## B.1 PSEUDO-CODE

---

**Algorithm 1:** HAG Algorithm

---

**Input:** Reservoir weights $\mathbf{W}$, Input weights $\mathbf{W}_{\text{in}}$, Bias $\mathbf{b}$, Pretraining data $X_{\text{pretrain}}$,
Hyperparameters $(\rho, \beta, \delta w, \gamma, T_{\min}, T_{\max})$
**Output:** Adjusted reservoir weights $\mathbf{W}$
**for** *each time increment $T_{current}$ sampled from logspace between $T_{min}$ and $T_{max}$* **do**
    **for** $t \leftarrow 1$ **to** $T_{current}$ **do**
        Update reservoir states:
        $\mathbf{x}[t+1] \leftarrow \sigma\left(\mathbf{W}\mathbf{x}[t] + \mathbf{W}_{\text{in}}\mathbf{u}[t] + \mathbf{b}\right)$
    **end**
    **for** *each neuron $i$* **do**
        Compute activity measure $s_i$ (mean or variance over $T_{\text{current}}$)
        Compute growth indicator:
        $\Delta z_i \leftarrow \dfrac{1}{\beta}\left(s_i - \rho\right)$
        **if** $\Delta z_i < -1$ **then**
            Find neuron $j$ with highest Pearson correlation with neuron $i$
            Increase weight:
            $w_{ij} \leftarrow w_{ij} + \delta w$
            **if** *degree(i) ¿ $\gamma$* **then**
                // Implement logic to maintain maximum degree
            **end**
        **end**
        **if** $\Delta z_i > +1$ **then**
            Randomly select a synapse $w_{ij}$ connected to neuron $i$ to decrease
            $w_{ij} \leftarrow \max(w_{ij} - \delta w, 0)$
                          `// Ensure non-negativity`
        **end**
        **if** *variance-HAG and $x_i$ exceeds saturation threshold $\theta_{sat}$* **then**
            **for** *each outgoing synapse $w_{ij}$ of neuron $i$* **do**
                $w_{ij} \leftarrow w_{ij} \times \eta_{\text{sat}}$
            **end**
        **end**
    **end**
**end**

---

## B.2 CONNECTION FORMATION BASED ON PEARSON CORRELATION

**Identifying the Most Correlated Pair.** To form new connections, we consider only neurons $i$ and $j$ for which the growth indicator $\Delta z_i$ or $\Delta z_j$ satisfies:

$$\Delta z_i > 1 \quad \text{or} \quad \Delta z_j > 1. \tag{9}$$

For this subset of neurons, we compute pairwise correlation coefficients $r_{ij}$ for all pairs, with $r_{ij}$ as defined in A.1. The pair $(i^*, j^*)$ with the highest absolute correlation is selected:

$$(i^*, j^*) = \arg\max_{(i,j)} |r_{ij}|, \tag{10}$$

where the maximization is performed over all neuron pairs $(i, j)$ that have not yet achieved homeostasis.

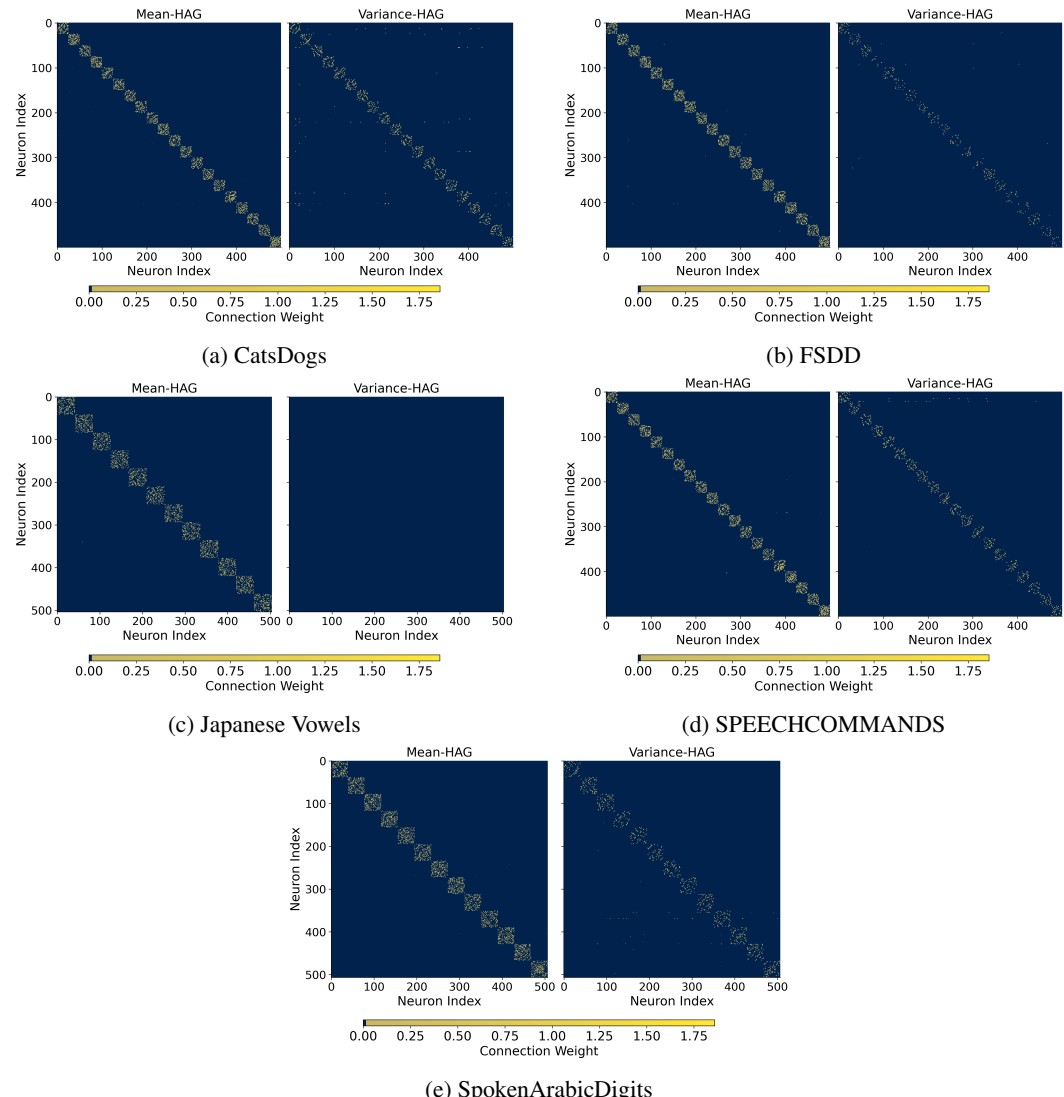

Figure 7: Final connectivity matrices generated using Mean-HAG and Variance-HAG algorithms across different classification datasets. For each dataset, the left panel represents the Mean-HAG connectivity matrix, while the right panel illustrates the Variance-HAG connectivity matrix.

**Establishing the Connection.** Once the most highly correlated pair $(i^*, j^*)$ is identified, a connection is established between these two neurons by incrementing the corresponding weight $w_{i^* j^*}$ in the connectivity matrix $\mathbf{W}$. The updated weight is given by:

$$w_{i^* j^*} \leftarrow w_{i^* j^*} + \delta w,$$

where $\delta w > 0$ is the weight increment parameter.

This mechanism ensures that connections are formed preferentially between neurons that exhibit high correlation, promoting the restructuring of the reservoir to enhance its dynamic representation of input data.

### B.3 FINAL CONNECTIVITY DETAILS

This section provides an analysis of the connectivity matrices produced by the Hebbian Architecture Generation (HAG) mechanism, which dynamically adjusts the reservoir's synaptic connections to

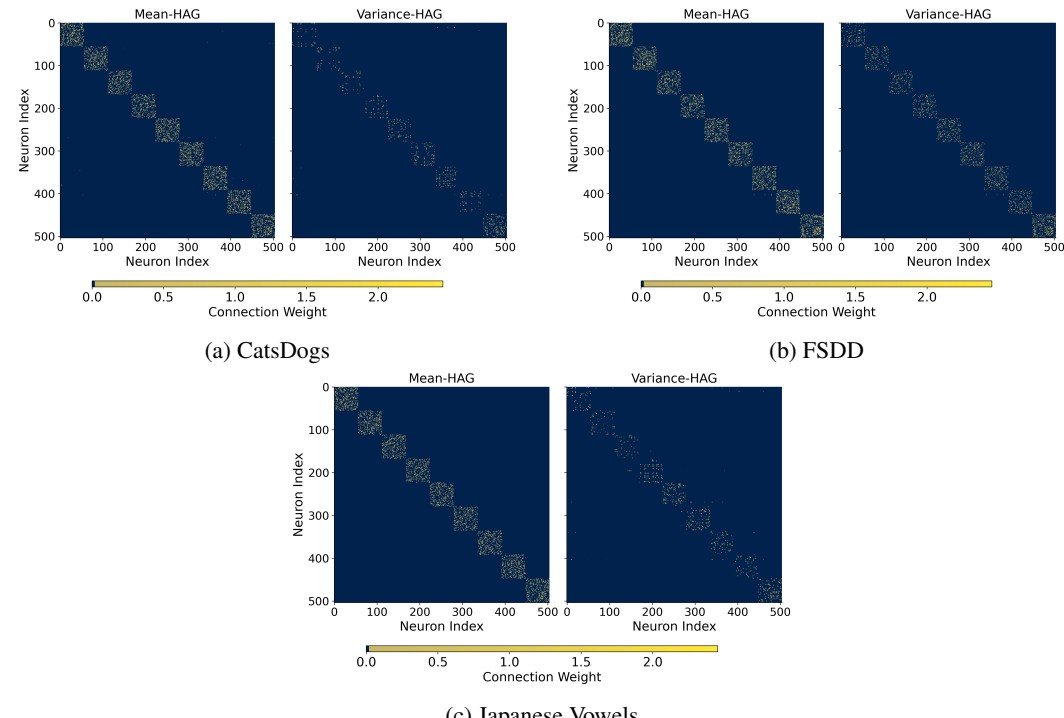

(a) CatsDogs

(b) FSDD

(c) Japanese Vowels

Figure 8: Final connectivity matrices generated using Mean-HAG and Variance-HAG algorithms across different prediction datasets. For each dataset, the left panel represents the Mean-HAG connectivity matrix, while the right panel illustrates the Variance-HAG connectivity matrix.

optimize its representational capacity. The visualizations in Figure 7 and Figure 8 reveal distinct patterns in the matrices depending on the dataset and the specific variant of the HAG algorithm used.

The connectivity matrices exhibit two notable characteristics. First, Mean-HAG generates structured, modular patterns indicative of clusters of strongly interconnected neurons. This structure corresponds to localized groups of neurons that represent specific features of the input space, reinforcing separability within subsets of the data. By contrast, Variance-HAG produces more dispersed and heterogeneous patterns, reflecting its goal of enhancing variability and reducing redundancy across neuron activations. Second, there is a distinct set of connections corresponding to inter-cluster connectivity, which increases the overall decorrelation of the reservoir state by linking the most redundant subsets.

The matrices also reveal that HAG adapts to dataset complexity. For instance, tasks involving high-dimensional data, such as *CatsDogs* and *SPEECHCOMMANDS*, result in connectivity matrices with pronounced modularity or hierarchical structure, emphasizing separability and diversity. However, the best-performing network (variance-HAG) is the one that exhibits the most inter-cluster connectivity.

On *Japanese Vowels*, HAG shows ill-conditioning as no connections were created. This explains why results on this dataset were so low, as the reservoir part of the network remained inadequately configured for effective processing.

# C  DETAILS ON THE HYPERPARAMETER OPTIMIZATION

In this appendix, we present the detailed results of the hyperparameter optimization across the different datasets. The hyperparameters were optimized using Optuna over 400 trials per dataset.

Table 5: Hyperparameter ranges, settings, and notation

| CATEGORY | PARAMETER | SYMBOL | RANGE/SETTINGS |
|---|---|---|---|
| **Shared Across Models** | Input Scaling | $s_{\text{in}}$ | 0.01 to 0.2, step 0.005 |
| | Bias Scaling | $s_b$ | 0 to 0.2, step 0.005 |
| | Ridge Coefficient | $\lambda$ | Logarithmic scale from $10^{-15}$ to $10^1$ |
| | Maximum degree | $\gamma$ | Between 10 to 20 |
| | Weight Increment | $\delta w$ | 0.001 to 0.1, step 0.001 |
| **mean-HAG** | Rate Target | $\rho_r$ | 0.5 to 1, step 0.01 |
| | Rate Spread | $\beta_r$ | 0.01 to 0.4, step 0.005 |
| **variance-HAG** | Variance Target | $\rho_v$ | 0.001 to 0.02, step 0.001 |
| | Variance Spread | $\beta_v$ | 0.001 to 0.02, step 0.001 |
| | Saturation Threshold | $\theta_{\text{sat}}$ | 0.8 to 0.98, step 0.02 |
| | Saturation scaling | $\eta_{\text{sat}}$ | 0.8 to 0.98, step 0.02 |
| | Connection Selection | - | "random" or "pearson" |
| **E-ESN/ESN** | Connectivity | $p$ | 0 to 1, no step |
| | Spectral Radius | $\rho_s$ | 0.4 to 1.6, step 0.01 |
| **IP-ESN** | Target distribution's mean | $\mu$ | 0 to 1, no step |
| | Target distribution's variance | $\sigma_{\text{IP}}$ | 0 to 1, no step |
| **Fixed Parameters** | Activation Function | $\sigma$ | Hyperbolic Tangent (tanh) |
| | $W_{in}$ Connectivity | - | 1 (Full Connectivity) |
| | Network Size | $n$ | Just above 500 neurons |
| | Input Duplication | - | Equal duplication for each input variate |

Table 6: Optimized hyperparameters for the variance-HAG algorithm

| | $s_{\text{in}}$ | $s_b$ | $\lambda$ | $\rho_v$ | $\beta_v$ | $\theta_{\text{sat}}$ | $\eta_{\text{sat}}$ | $\delta w$ | $\gamma$ |
|---|---|---|---|---|---|---|---|---|---|
| CatsDogs | 0.030 | 0.000 | $10^0$ | 0.017 | 0.007 | 0.86 | 0.88 | 0.100 | 11 |
| FSDD | 0.155 | 0.020 | $10^{-7}$ | 0.011 | 0.001 | 0.82 | 0.94 | 0.087 | 14 |
| Japanese Vowels | 0.040 | 0.190 | $10^{-9}$ | 0.003 | 0.013 | 0.90 | 0.86 | 0.075 | 18 |
| Lorenz | 0.065 | 0.170 | $10^{-9}$ | 0.018 | 0.005 | 0.94 | 0.96 | 0.006 | 17 |
| MackeyGlass | 0.170 | 0.185 | $10^{-9}$ | 0.019 | 0.007 | 0.80 | 0.94 | 0.076 | 10 |
| SPEECHCOMMANDS | 0.060 | 0.000 | $10^{-9}$ | 0.014 | 0.003 | 0.92 | 0.80 | 0.030 | 20 |
| SpokenArabicDigits | 0.060 | 0.040 | $10^{-10}$ | 0.014 | 0.001 | 0.82 | 0.96 | 0.057 | 16 |
| Sunspot | 0.180 | 0.135 | $10^{-9}$ | 0.017 | 0.005 | 0.94 | 0.94 | 0.028 | 14 |

The detailed hyperparameter settings serve as a reference for reproducing the results and offer insights into how different parameters impact the performance of the different algorithm.

## C.1 HYPERPARAMETER DEFINITIONS:

- **Input Scaling ($s_{\text{in}}$):** Scaling factor applied to the input weights $\mathbf{W}_{\text{in}}$.

Table 7: Optimized hyperparameters for the mean-HAG algorithm

|  | $s_{\text{in}}$ | $s_b$ | $\lambda$ | $\rho_r$ | $\beta_r$ | $\delta w$ | $\gamma$ |
|---|---|---|---|---|---|---|---|
| CatsDogs | 0.120 | 0.100 | $10^0$ | 0.51 | 0.39 | 0.021 | 12 |
| FSDD | 0.200 | 0.075 | $10^{-5}$ | 0.56 | 0.39 | 0.050 | 10 |
| Japanese Vowels | 0.195 | 0.195 | $10^{-4}$ | 0.66 | 0.21 | 0.094 | 13 |
| Lorenz | 0.190 | 0.170 | $10^{-9}$ | 0.97 | 0.165 | 0.021 | 19 |
| MackeyGlass | 0.045 | 0.185 | $10^{-10}$ | 0.94 | 0.295 | 0.043 | 10 |
| SPEECHCOMMANDS | 0.060 | 0.000 | $10^{-9}$ | 0.58 | 0.34 | 0.002 | 16 |
| SpokenArabicDigits | 0.075 | 0.035 | $10^{-10}$ | 0.53 | 0.38 | 0.025 | 11 |
| Sunspot | 0.050 | 0.105 | $10^{-9}$ | 0.51 | 0.265 | 0.087 | 13 |

Table 8: Optimized hyperparameters for E-ESN

|  | $s_{\text{in}}$ | $s_b$ | $\lambda$ | $p$ | $\rho_s$ |
|---|---|---|---|---|---|
| CatsDogs | 0.135 | 0.025 | $10^{-6}$ | 0.842 | 0.49 |
| FSDD | 0.175 | 0.14 | $10^{-9}$ | 0.0040 | 0.92 |
| Japanese Vowels | 0.190 | 0.005 | $10^{-9}$ | 0.155 | 0.62 |
| MackeyGlass | 0.195 | 0.2 | $10^{-10}$ | 0.0318 | 1.4 |
| SPEECHCOMMANDS | 0.040 | 0.05 | $10^{-12}$ | 0.0038 | 1.02 |
| SpokenArabicDigits | 0.030 | 0.1 | $10^{-14}$ | 0.0049 | 1.0 |
| Sunspot | 0.055 | 0.005 | $10^{-9}$ | 0.0178 | 0.78 |

- **Bias Scaling** ($s_b$)**:** Scaling factor applied to the bias vector $\mathbf{b}$.

- **Ridge Coefficient** ($\lambda$)**:** Regularization parameter in ridge regression, where $\lambda = 10^{\text{ridge exponent}}$.

- **Variance Target** ($\rho_v$)**:** Target standard deviation of neuron states.

- **Rate Spread** ($\beta_r$)**:** Spread parameter activity controlling the activity deviation from target $\rho_r$ that is tolerated.

- **Rate Target** ($\rho_r$)**:** Target activity of neuron states.

- **Variance Spread** ($\beta_v$)**:** Spread parameter controlling the sensitivity to deviations from $\rho_v$.

- **Saturation Threshold** ($\theta_{\text{sat}}$)**:** Threshold beyond which intrinsic plasticity mechanisms reduce synaptic weights.

- **Saturation scaling** ($\eta_{\text{sat}}$)**:** Factor by which synaptic weights are scaled when saturation occurs.

- **Weight Increment** ($\delta w$)**:** Amount by which synaptic weights are increased during growth.

- **Maximum Degree** ($\gamma$)**:** Maximum number of synaptic partners per neuron.

- **Mean of the target distribution for IP** ($\mu$)**:** Target mean for intrinsic plasticity normalization.

- **Variance of the target distribution for IP** ($\sigma_{\text{IP}}$)**:** Target standard deviation for intrinsic plasticity normalization.

## C.2 OPTIMIZED HYPERPARAMETERS FOR EACH ALGORITHM

Tables 7, 6, 8, 9 and 10 summarize the optimal hyperparameters found for each dataset and algorithm. Key insights include:

1. **Variance-HAG and Mean-HAG:** Effective in dynamically tailoring the reservoir to the data, with significant reliance on $\rho_v$, $\rho_r$, and $\gamma$ to optimize neuron activity and connectivity.

Table 9: Optimized hyperparameters for traditional ESN

|  | $s_{\text{in}}$ | $s_b$ | $\lambda$ | $p$ | $\rho_s$ |
|---|---|---|---|---|---|
| CatsDogs | 0.135 | 0.025 | $10^{-6}$ | 0.842 | 0.49 |
| FSDD | 0.175 | 0.140 | $10^{-9}$ | 0.004 | 0.92 |
| Japanese Vowels | 0.190 | 0.005 | $10^{-9}$ | 0.155 | 0.62 |
| MackeyGlass | 0.130 | 0.045 | $10^{-10}$ | 0.193 | 0.89 |
| SPEECHCOMMANDS | 0.135 | 0.005 | $10^{-11}$ | 0.356 | 1.00 |
| SpokenArabicDigits | 0.035 | 0.005 | $10^{-14}$ | 0.003 | 0.61 |
| Sunspot | 0.025 | 0.110 | $10^{-11}$ | 0.014 | 0.46 |

2. **Traditional ESN and E-ESN:** Spectral radius ($\rho_s$) and connectivity ($p$) are critical for matching the reservoir's memory and dynamic properties to the dataset.

3. **General Trends:** High input scaling ($s_{\text{in}}$) and low ridge coefficients ($\lambda$) are commonly effective across algorithms, reflecting the need for strong input signals and minimal regularization.

Table 10: Optimized hyperparameters for IP-ESN

|  | $s_{\text{in}}$ | $s_b$ | $\lambda$ | $p$ | $\rho_s$ | $\mu$ | $\sigma$ |
|---|---|---|---|---|---|---|---|
| CatsDogs | 0.18 | 0.08 | $10^{-10}$ | 0.462 | 0.46 | 0.733 | 0.142 |
| FSDD | 0.05 | 0.095 | $10^{-14}$ | 0.005 | 1.05 | 0.298 | 0.087 |
| Japanese Vowels | 0.14 | 0.06 | $10^{-10}$ | 0.327 | 0.56 | 0.886 | 0.719 |
| Lorenz | 0.09 | 0.06 | $10^{-9}$ | 0.093 | 1.11 | 0.071 | 0.838 |
| MackeyGlass | 0.19 | 0.19 | $10^{-7}$ | 0.048 | 0.45 | 0.158 | 0.363 |
| SPEECHCOMMANDS | 0.195 | 0.105 | $10^{-14}$ | 0.005 | 0.89 | 0.097 | 0.062 |
| SpokenArabicDigits | 0.03 | 0.125 | $10^{-15}$ | 0.015 | 0.82 | 0.318 | 0.582 |
| Sunspot | 0.01 | 0.145 | $10^{-11}$ | 0.138 | 0.64 | 0.758 | 0.782 |

# D  ADDITIONAL RESULTS AND ANALYSIS

To further substantiate the effectiveness of our Hebbian Architecture Generation (HAG) method, we have expanded our experiments to include detailed results and additional metrics across our datasets and reservoir configurations. Specifically, we provide an in-depth analysis of the spectral radius, Pearson correlation coefficients among neuron activations, and the Cumulative Explained Variance (CEV) in the reservoir states. These metrics offer insights into the dynamical properties of the reservoirs and their impact on performance. Table 11 and 12 summarizes the spectral radius, average Pearson correlation, and CEV for each combination of dataset and reservoir configuration. The spectral radius is measured from the connectivity matrix of the different networks obtained with various rules, while Pearson correlation and CEV are measured based on neurons' activity during the test set inference for each dataset.

## D.1  RESULTS

### D.1.1  SPECTRAL RADII

The spectral radius is an important parameter in Echo State Networks (ESNs), influencing the echo state property. Our results indicate that the HAG-based methods (*Mean-HAG* and *Variance-HAG*) generally have higher spectral radii compared to the traditional E-ESN, particularly in datasets like SPEECHCOMMANDS and CatsDogs (see Figure 9a). This suggests that the adaptive connectivity in HAG methods allows the reservoir to have a high spectral radius while performing extremly well, for instance the SPEECHCOMMANDS dataset.

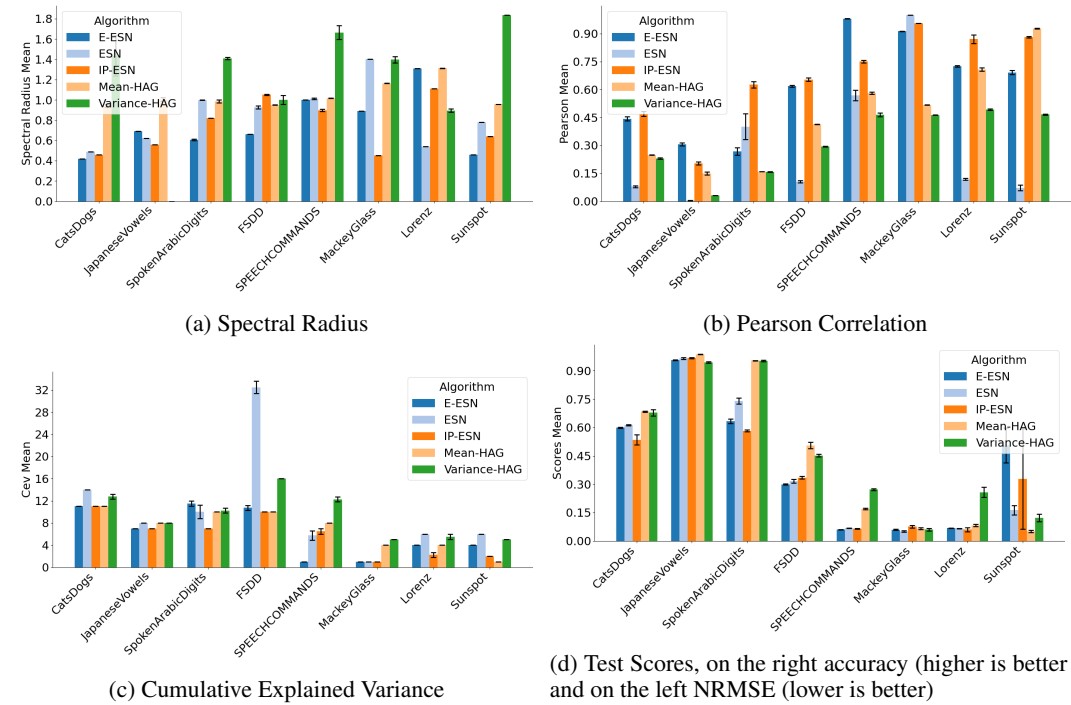

(a) Spectral Radius

(b) Pearson Correlation

(c) Cumulative Explained Variance

(d) Test Scores, on the right accuracy (higher is better) and on the left NRMSE (lower is better)

Figure 9: Detailed Results for Spectral Radius, Pearson Correlation, Cumulative Explained Variance, and Test Scores across different reservoir configurations and Datasets. Each subfigure illustrates the respective metric for various algorithms across the evaluated datasets.

### D.1.2 PEARSON CORRELATION

The average Pearson correlation coefficient among neuron activations provides insight into the redundancy and diversity of the reservoir states. Lower correlation values imply a more decorrelated and thus more informative set of features for the readout layer.

In our experiments, the *Variance-HAG* method consistently achieved lower Pearson correlation coefficients compared to other methods across multiple datasets, as shown in Table 11 and Table 12 and illustrated in Figure 9b. For instance, in the SPEECHCOMMANDS dataset, *Variance-HAG* achieved a correlation of $0.471$, significantly lower than the $0.980$ observed in the random E-ESN. Similarly, in the MackeyGlass dataset, *Variance-HAG* achieved a correlation of $0.064$, compared to $0.910$ in the random E-ESN. This reduction in correlation indicates that *Variance-HAG* effectively decorrelates neuron activations, potentially leading to better generalization and performance.

### D.1.3 CUMULATIVE EXPLAINED VARIANCE

The Cumulative Explained Variance (CEV) quantifies the dimensionality of the reservoir's projected feature space by identifying the number of principal components needed to explain a specified fraction of the variance. Higher CEV values indicate that more components are required to capture the system's dynamics, which can reflect richer dynamics or increased complexity.

Our results show that the HAG methods doesn't always exhibit higher CEV values. But it is important to notice that it does on the most complex dataset (SPEECHCOMMAND) where our algorithm shows the most impressive results.

### D.2 DISCUSSION

The analysis of these metrics supports the efficacy of the HAG methods in improving reservoir performance:

Table 11: Spectral radius, Pearson Correlation, Cumulative Explained Variance (CEV), and test scores for the different reservoir configurations across prediction datasets' test data (average over 4 trials).

| Dataset | Algorithm | SR | Correlation | CEV | Test Scores |
|---|---|---|---|---|---|
| | E-ESN | $0.89 \pm 0.00$ | $0.913 \pm 0.001$ | $1.0 \pm 0.0$ | $0.059 \pm 0.004$ |
| | ESN | $1.40 \pm 0.00$ | $1.00 \pm 0.00$ | $1.0 \pm 0.0$ | $0.051 \pm 0.005$ |
| MackeyGlass | IP-ESN | $0.45 \pm 0.00$ | $0.955 \pm 0.001$ | $1.0 \pm 0.0$ | $0.076 \pm 0.007$ |
| | Mean-HAG | $1.16 \pm 0.00$ | $0.517 \pm 0.001$ | $4.0 \pm 0.0$ | $0.065 \pm 0.005$ |
| | Variance-HAG | $1.40 \pm 0.03$ | $0.463 \pm 0.001$ | $5.0 \pm 0.0$ | $0.060 \pm 0.007$ |
| | E-ESN | $1.31 \pm 0.00$ | $0.725 \pm 0.004$ | $4.0 \pm 0.0$ | $0.068 \pm 0.001$ |
| | ESN | $0.54 \pm 0.00$ | $0.118 \pm 0.005$ | $6.0 \pm 0.0$ | $0.066 \pm 0.001$ |
| Lorenz | IP-ESN | $1.11 \pm 0.00$ | $0.871 \pm 0.023$ | $2.3 \pm 0.4$ | $0.060 \pm 0.011$ |
| | Mean-HAG | $1.31 \pm 0.00$ | $0.707 \pm 0.009$ | $4.0 \pm 0.0$ | $0.083 \pm 0.006$ |
| | Variance-HAG | $0.90 \pm 0.02$ | $0.493 \pm 0.004$ | $5.5 \pm 0.5$ | $0.257 \pm 0.027$ |
| | E-ESN | $0.46 \pm 0.00$ | $0.691 \pm 0.011$ | $4.0 \pm 0.0$ | $0.505 \pm 0.092$ |
| | ESN | $0.78 \pm 0.00$ | $0.072 \pm 0.014$ | $6.0 \pm 0.0$ | $0.163 \pm 0.024$ |
| Sunspot | IP-ESN | $0.64 \pm 0.00$ | $0.882 \pm 0.004$ | $2.0 \pm 0.0$ | $0.610 \pm 0.687$ |
| | Mean-HAG | $0.96 \pm 0.00$ | $0.928 \pm 0.002$ | $1.0 \pm 0.0$ | $0.051 \pm 0.005$ |
| | Variance-HAG | $1.84 \pm 0.00$ | $0.466 \pm 0.002$ | $5.0 \pm 0.0$ | $0.122 \pm 0.020$ |

- **Enhanced Dynamics**: Higher spectral radii in HAG methods suggest a more powerful dynamic regime, allowing the reservoir to better model temporal dependencies.

- **Reduced Redundancy**: Lower Pearson correlations indicate that HAG methods produce more diverse neuron activations, reducing redundancy and providing richer information to the readout layer.

- **Enriched Feature Space**: Higher CEV values demonstrate that HAG methods generate a more informative and expansive feature space, facilitating better representation of input dynamics.

However, while HAG algorithms excel in classification tasks by enhancing feature separability, their performance in prediction tasks is more variable. As shown in Table 11 and Table 12, *Variance-HAG* does not always outperform traditional ESNs or other baseline models in prediction scenarios. This inconsistency suggests that the mechanisms driving connectivity adjustments in HAG may be more aligned with tasks requiring distinct feature separation rather than continuous temporal forecasting.

### D.3 IMPLICATIONS FOR RESERVOIR DESIGN

Despite these shortcomings, the HAG methods offer valuable insights into how adaptive connectivity can enhance reservoir performance. The ability to dynamically tailor the reservoir to specific tasks by reducing neuron correlations and expanding the feature space underscores the potential of biologically inspired design principles in neural network architecture.

It is important to note that enriching the feature space (lower Pearson correlation and higher CEV) does not always translate into augmented performances across all tasks. This discrepancy may stem from the inherent differences between classification and prediction tasks. Classification tasks primarily benefit from enhanced feature separability, allowing for more accurate differentiation between classes. In contrast, prediction tasks rely heavily on the reservoir's ability to capture and retain temporal dependencies, which may not be directly enhanced by the structural adjustments made by HAG.

In conclusion, the Hebbian Architecture Generation (HAG) method presents a robust framework for enhancing reservoir computing, particularly in classification tasks, by leveraging biologically inspired adaptive connectivity. While its efficacy in prediction tasks is promising, it highlights the need for a nuanced approach that considers the distinct demands of different task types. The

Table 12: Spectral radius, Pearson Correlation, Cumulative Explained Variance (CEV), and test scores for the different reservoir configurations across classification datasets' test data (average over 4 trials).

| Dataset | Algorithm | SR | Correlation | CEV | Test Scores |
|---|---|---|---|---|---|
| CatsDogs | E-ESN | $0.42 \pm 0.00$ | $0.443 \pm 0.011$ | $11.0 \pm 0.0$ | $0.599 \pm 0.003$ |
| | ESN | $0.49 \pm 0.00$ | $0.078 \pm 0.005$ | $14.0 \pm 0.0$ | $0.613 \pm 0.003$ |
| | IP-ESN | $0.46 \pm 0.00$ | $0.469 \pm 0.013$ | $11.0 \pm 0.0$ | $0.535 \pm 0.026$ |
| | Mean-HAG | $0.94 \pm 0.02$ | $0.249 \pm 0.001$ | $11.0 \pm 0.0$ | $0.684 \pm 0.003$ |
| | Variance-HAG | $1.44 \pm 0.20$ | $0.230 \pm 0.004$ | $12.75 \pm 0.43$ | $0.678 \pm 0.017$ |
| JapaneseVowels | E-ESN | $0.69 \pm 0.00$ | $0.306 \pm 0.008$ | $7.0 \pm 0.0$ | $0.957 \pm 0.002$ |
| | ESN | $0.62 \pm 0.00$ | $0.004 \pm 0.000$ | $8.0 \pm 0.0$ | $0.966 \pm 0.005$ |
| | IP-ESN | $0.56 \pm 0.00$ | $0.203 \pm 0.008$ | $7.0 \pm 0.0$ | $0.968 \pm 0.003$ |
| | Mean-HAG | $1.00 \pm 0.02$ | $0.149 \pm 0.007$ | $8.0 \pm 0.0$ | $0.987 \pm 0.001$ |
| | Variance-HAG | $0.00 \pm 0.00$ | $0.031 \pm 0.000$ | $8.0 \pm 0.0$ | $0.945 \pm 0.003$ |
| SpokenArabicDigits | E-ESN | $0.61 \pm 0.01$ | $0.268 \pm 0.020$ | $11.5 \pm 0.5$ | $0.634 \pm 0.012$ |
| | ESN | $1.00 \pm 0.00$ | $0.401 \pm 0.068$ | $10.0 \pm 1.22$ | $0.740 \pm 0.016$ |
| | IP-ESN | $0.82 \pm 0.00$ | $0.626 \pm 0.017$ | $7.0 \pm 0.0$ | $0.582 \pm 0.005$ |
| | Mean-HAG | $0.99 \pm 0.01$ | $0.159 \pm 0.000$ | $10.0 \pm 0.0$ | $0.954 \pm 0.002$ |
| | Variance-HAG | $1.41 \pm 0.01$ | $0.157 \pm 0.003$ | $10.25 \pm 0.43$ | $0.952 \pm 0.004$ |
| FSDD | E-ESN | $0.66 \pm 0.00$ | $0.618 \pm 0.005$ | $10.75 \pm 0.43$ | $0.299 \pm 0.004$ |
| | ESN | $0.93 \pm 0.01$ | $0.105 \pm 0.006$ | $32.5 \pm 1.12$ | $0.316 \pm 0.010$ |
| | IP-ESN | $1.05 \pm 0.01$ | $0.653 \pm 0.009$ | $10.0 \pm 0.0$ | $0.335 \pm 0.007$ |
| | Mean-HAG | $0.95 \pm 0.00$ | $0.414 \pm 0.001$ | $10.0 \pm 0.0$ | $0.506 \pm 0.017$ |
| | Variance-HAG | $1.00 \pm 0.04$ | $0.293 \pm 0.002$ | $16.0 \pm 0.0$ | $0.452 \pm 0.006$ |
| SPEECHCOMMANDS | E-ESN | $1.00 \pm 0.00$ | $0.980 \pm 0.001$ | $1.0 \pm 0.0$ | $0.060 \pm 0.001$ |
| | ESN | $1.01 \pm 0.01$ | $0.569 \pm 0.028$ | $5.75 \pm 0.83$ | $0.068 \pm 0.001$ |
| | IP-ESN | $0.90 \pm 0.01$ | $0.750 \pm 0.007$ | $6.5 \pm 0.5$ | $0.065 \pm 0.002$ |
| | Mean-HAG | $1.02 \pm 0.00$ | $0.581 \pm 0.005$ | $8.0 \pm 0.0$ | $0.169 \pm 0.004$ |
| | Variance-HAG | $1.66 \pm 0.07$ | $0.464 \pm 0.009$ | $12.25 \pm 0.43$ | $0.272 \pm 0.006$ |

adaptability and biological plausibility of HAG not only address the limitations of static reservoir architectures but also pave the way for more versatile and efficient neural network models in diverse learning environments.

