# OpenReview forum: "Reshaping Reservoirs: Hebbian Plasticity for Improved Data Separability"
_ICLR.cc/2025/Conference — ICLR 2025 Conference Withdrawn Submission_

### Official Review · Reviewer_tQ4Q · 2024-10-30

**Soundness:** 2
**Presentation:** 3
**Contribution:** 2
**Rating:** 6
**Confidence:** 4

**Summary:**

This paper suggests a method for self-organizing reservoirs that obey a mixture of a hebbian rule as well as a mechanism to grow connections, which is a function of pearson correlations between neurons outside of homeostatis; and pruning, which is random. The objective of these adaptive mechanisms is to de-correlate features within a reservoir in order to allow for better (linearly seperable) projections. The architecture that is used is an RNN of approximately 500 nodes (depending on the task). Across a variety of tasks, including several audio classification tasks and complex-system time-series prediction tasks, the authors showed that their methods (Hebbian Architecture Generation) HAG achieved consistently better performance when compared to standard echo state networks.

**Strengths:**

Originality: While many of the components of the methods used in this paper have been used elsewhere, I believe it is original to combine them with reservoir computers, and in this sense, this is an original implementation of some known methods.

Quality: The authors did well to select a broad set of tasks, and especially some difficult tasks, taking care to ensure performance scores and training were done with care by implementing appropriate cross-validation techniques.

Clarity: The goals and intentions of this project are sound and well motivated. Reservoirs are, by their nature, extremely sensitive to their initializations, and there are a number of papers that discuss clever ways to initialize these models in order to have them behave reasonably well across different tasks. It is likely that no single initialization technique would work for all tasks, and so it makes sense that an adaptive method, inspired by neural plasticity would be a step in the right direction.

Significance: I believe the project has the potential to be signficant as methods for self-organized/adaptive unsupervised networks are still not well understood and by and large do not have the same performances as more standard machine learning practices. Discovering new and efficient methods to modify architectures online, I believe, will be important in understanding and implementing biologically inspired learning algorithms in the future. This project's implementation explores a reasonable mechanism for architecture modifications online. However, this signifiance is slightly overshadowed by some lack of detail, which I try to outline below.

**Weaknesses:**

To my understanding, the novel contribution of this paper is largely contained in the growth/pruning mechanism introduced. The hebbian rule itself looks to me to be relatively standard (though some details are still unclear to me, for example in deciding exactly *which* connection is strengthened when $\Delta z < -1$), so the biggest difference is in using Pearson correlations to decide when new connections form. However, there is not much information in the rest of the paper of the dynamics of this rule over time. What effect does this have on the architecture? Are there clusters in the end? What kind of connectivity does it result in? What is the difference between the spectral radius before and after the mechanism? If this is the novel contribution of the paper, I would have expected quite a bit more detail on the effect this mechanism has on the architecture, but the only result that is shown is in figure 5 and 6 and I'm not sure if figure 6 is telling the story the paper claims it does (see below on more comments on figure 6).

### Edit (30/10/2024)
edit: Looking further into it, I realize that actually the plasticity rule that is being used is not hebbian at all, it's actually a homeostatic rule. If I'm not mistaken, a hebbian rule involves some type of product of activations, whereas the rule here is just trying to match a target rate for each individual neuron? Is it not incorrect then to call this a hebbian rule?

Furthermore, the usage of pearson correlations for all neurons very explicitly makes this rule global both in space (nodes) and time, as computing the correlation values requires a large enough timeseries. So I have some problems with how this rule is being pitched/described. It's neither exactly hebbian nor local. I think it ought to be described more accurately and acknowledging these global elements.

# Some general comments:
- The paper would benefit greatly from an appendix which delves more into the detail of the methodology used and visualized some of the dynamics of the self-organizing principles.
    - how does the pruning/growing mechanism affect the architecture of the network over time? How significant are the differences to the random networks? To a network with the hebbian rule without the pruning/growing? At the moment I have no idea how much this mechanism, which I have to say is really the only novel part of the model as far as I understand, is actually benefiting the reservoir. Your performance scores are alright, but some of them are hardly different from the ESN. You have one score that is significantly better for the SPEECHCOMMANDS tasks. Is it possible that the growth/pruning is doing more there? Can you quantify how much this is doing?
    - an ablation analysis, if you remove the growth/pruning, do your results diminish? Is it important?
    - after your hyper-parameter search, what values were successful? Is there a heat-map that you can make for the hyperparameters to show what worked and what didn't? If I were to try to recreate your results I would have no idea what results you had to even compare to!
- the figures in the paper are on the very edge of legible and need their sizes and fontsizes standerdized. Please look up the page widths of the paper and make sure your figures have the right sizes and dpi settings. (In matplotlib you would set the figsize=( ) parameter.)
    - Also, the linewidths, axis labels and legends need some work. For example figure 3, the left and right-most panels have their y-axis as "Value". That is not an informative label.
    - Same for figure 2 and 4.
    - Figure 2, though somewhat useful to see (in the appendix) is taking up space that I would rather be used to explain the growth/pruning mechanism.
    - Figure 2 also has some weird text at the 0 mark which is illegible. The legend is also hardly noticably different between the different shades. But I think you should remove this figure anyway or put in appendix.

# Typos and unclear things
- the values $\Delta w$ and $\delta w$ (why are they different?) are not clear to me (is it the "weight increment" in the table? Actually, all variables in the table that have already been introduced in the paper should probably include their symbols.)
- line 165 typo: "we then identifying highly correlated neuron pairs we establishes a connction" needs rewriting
- line 166 typo: "if $\Delta z_i > +1$ connections weight" should probably read "if $\Delta z_i > +1$ the connection weight needs to be..."
- line 172: "maximum number of synaptic partners, $\gamma$" isn't this just the degree of the node? It would probably be more clear if it was written that way
- line 174 typo: "which might limits the..." should be "which might limit the"
- line 297 typo: "actual targetst"
- there were a few more small typos so I won't list them all, but the paper need some editing still

# References (edit 30/10/2024)
There are a number of extremely relevant references and works that I think this project doesn't cite that it should.
- [Homeostatic Plasticity and External Input Shape Neural Network Dynamics](https://journals.aps.org/prx/abstract/10.1103/PhysRevX.8.031018)
    - As this is largely a homeostatic rule and not a hebbian one, I was surprised to not find this work in the citations.
- [SORN: a self-organizing recurrent neural network](https://www.frontiersin.org/journals/computational-neuroscience/articles/10.3389/neuro.10.023.2009/full)
    - similarly, another project with very similar goals is described here, and although they use inhibitory neurons as well, it is too relevant not to acknowledge.


Overall: I would say this paper could use another iteration of development. It is currently lacking the detail for someone to recreate the work, firstly. And secondly, considering that it is utilizing techniques that have been used before and are relatively simple (this is a good thing), and that it is doing so in a relatively small increment (again, this is not bad), this requires that this increment be well justified and argued for. At the moment, it is not at all clear to me how the components of this metholodgy contribute to the final result. We only see the final result as a table of performances, but it is not clear to me in what way the adaptive mechanisms contributed to this. This needs to be made clear, and the mechanisms need to be understood in their role for changing the architecture and clustering correlated features.

**Questions:**

I made a list of questions in the Weaknesses section (regarding stuff I would have liked to see in the appendix for example that I think are required for me to know for sure that the methods introduced in this paper are actually responsible for the results presented, as well as for reproducability reasons).

Some more questions I have that are less crucial but have piqued my curiousity are:
- How does performance scale with model size? Some of the results compared with the ESNs are not too different, and some are quite a bit better. If you scale the model to smaller or larger networks, are these differences conserved? Does your model adapt better or worse when the size changes?
- I think you compared your results to standard ESNs, and according to your table, there were a range of spectral radii that were used to initialize random networks. Which spectral radii did you use when making your performance tables? There is some literature on "best practices" for initializing reservoirs that specifically detail the sensitivity/performance of reservoirs on these settings. Did you use these "best practices" or were the ESNs handicapped by having random initializations? This part is not clear to me and unfortunately this is what needs to be made completely uncertain for the results of your method to be interesting.
- In figure 6, the evolution of the moving cumulative explained variance in panel (a) starts quite high, at around 22-25 and then decreases a bit down to ~19. Why does it start so high? In panel (b) the random reservoir starts around 2. If during training is when the growth/pruning happen, shouldn't panel (a) also start very low and then increase to higher principle components? If the growth/pruning is already done before making this figure, what was the point of this figure? I would be curious to know how the INITIAL network behaves and then how the growth/pruning + hebbian learning complexifies the dynamics. Right now it's not clear to me why figure 6 looks the way it does.
- line 102: "according to a self-organised local rule" I understand that the hebbian part of this mechanism is looking at individual neuron activity and in that sense it's local, but not much else is local in this model. As I understand it, the network architecture is fully connected (or connectable, so every neuron *could* in principle be connected to any other neuron) and also the growth/pruning mechanism is fully global as well. I would be careful when phrasing the rule as such, because actually there is a mixture of global and local interactions here.

---

> ### Author Response · Authors · 2024-11-22
> **Response to Reviewer Feedback - Weaknesses**
>
> I wanted to extend my deepest gratitude for your thorough and insightful review of our paper. This was extremely valuable to us. Below, we address your points systematically:
>
> Weaknesses:
>
> 1. Spectral radius : We appreciate your suggestion regarding the spectral radius. Given that the initial matrix starts as blank (with spectral radius 0), the differences emerge entirely due to the growth/pruning mechanism, limiting insights from the dynamics of spectral radius analysis. To address this, we have added a detailed analysis in Table 11 and Table 12 (Appendix D), which compares key metrics, including spectral radius.
> 2. Final architecture and pruning/growing effects: We are actively working on visualizations to better illustrate the dynamics of pruning and growing over time and its effects on the final architecture. We hope to include it in the revised version of the final manuscript before the end of the rebutal period. Indeed the final connectivity exhibits clustering, which we aim to show in the updated figures.
> 3. Local and Hebbian nature of the algorithm:
> * Your comment on the local nature of the rule is absolutely true, we remove it as indeed the rule involves global computations via Pearson correlations.
> * We have expanded the text to include a detailed explanation of the Pearson correlation calculation, highlighting how it represents a Hebbian learning rule (i.e., strengthening connections between neurons with correlated activity). In particular the rule involves as you noted « some type of product of activations » throughout the calculous of Pearson calculation. That might not have been clear from the first version of this paper as we didn’t include the formula for Pearson correlation, we hope that it will appear more evident now that the paper been expanded to include this detailed calculation. Additional details on the algorithm are available in Appendix B.
> 4. Comparison of the algorithms for each datasets  : To enhance interpretability, we have conducted further comparisons of our algorithm’s performance and metrics across datasets in appendix D. The performance improvements attributable to the HAG mechanism are now highlighted.
> 5. Final hyper-parameters values :
> * We have included moved the hyper parameter table in  (Table 5 in Appendix C)
> * To enhance the reproducibility of our work this Appendix C now include a listing of the successful hyper-parameter values identified through our search for each rule and datasets
> * Given the complexity and the extensive number of hyperparameters, providing detailed heatmaps for every optimization in the paper is impractical, but we are pleased to make the data available upon request for reproducibility purposes.
> 6. Figures improvements : We also acknowledge your remarks regarding the clarity and presentation of our figures. We have revise them to enhance readability and ensure they effectively convey the intended information. We also detailed the caption.
> * Figure 2: Figure 2 has been updated to better align with the section’s content. It now provides a clearer visualization of the role of \beta and \rho
> * Figure 3 and 4 have be more consistent and readable this include :  consistent axis labels and timescales across all subplots and an updated caption to clarify the purpose of each subplot
> * All figure's caption have been revised to enhance clarity.
> 7.  Typos : Additionally, we addresses all the typos and unclear statements you’ve pointed out and some others we found to improve the overall clarity of the manuscript.
> 8. References : Thank you also for bringing to our attention the relevant references (one of which Zierenberg2018 we didn’t know), we are actually fan on the work on SORN, we occluded to cite because it concerns spiking models of neurons but we shouldn’t have. We incorporated these works to better situate our research within the existing literature.
>
>
> We hope these revisions address your concerns and enhance the clarity, reproducibility, and impact of our work.

---

> ### Author Response · Authors · 2024-11-22
> **Response to Reviewer Feedback - Questions**
>
> We didn't had the space to reply to the question in our other comment since the number of character per answer is limited so we will complete it here.
>
> Questions:
>
> 1. Performance and model size: While we acknowledge the importance of scaling analyses, time constraints during the rebuttal period prevent us from including these results. This remains a valuable avenue for future work.
>
> 2. Initialization of the models : We used best practices to initialize the reservoir as found in 10.1016/j.cosrev.2009.03.005). This includes having random initialization with spectral radius and connectivity being optimize. For each task the spectral radius have been selected upon cross validation hyper parameters optimization, we added in this iteration an appendix to report the exact values and detailed a bit more the hyper parameters optimization there. For results on test data we use the best found hyper parameters.
>
> 3. Figure 6:  The high initial value in Figure 6(a) was due to the rolling window (500 timesteps) exceeding the initial growth/pruning interval. So you were right to point out that “growth/pruning is already done before making this figure.” The point of this figure was to show that the overall richness of the reservoir is improved by variance/mean-HAG compared to classic ESNs. However, your comment made us investigate or code more thoroughly and we actually found a bug that was miscalculating the exact value of CEVs. The current figures reflect this last change. We are still considering a clever way to visualize the transition period in the early start of the HAG methods.
>
> 4. As previously mentioned it was incorrect to describe the rule as « local » given that the Pearson correlation involves global computations so. We have removed this characterization from the manuscript and revised the text to accurately reflect the nature of the algorithm. Thank you again for pointing it out.
>
> We hope these revisions address your concerns and enhance the clarity, reproducibility, and impact of our work. Thank you again for your thoughtful feedback, which has been instrumental in improving our manuscript.

---

> ### Comment · Reviewer_tQ4Q · 2024-11-26
>
> I would like to thank the authors for the effort put into improving the paper and the additions they have made to answer some of the questions/concerns put worth here. I think the clarity of the paper has improved considerably, and for that I will increase my review score in presentation to 3 and to a "marginally above the acceptance threshold".
>
> I have to add, that I am still dubious a bit about the methods at large here, for the very reason that pearson correlations are globally calculated. I think this study may act as a good stepping stone for discovering more local plasticity methods, which I think is more in the spirit of reservoir computing (whose whole purpose is to not have to be optimized in an expensive way). Right now, this method is an interesting study into how decorrelating the system is helpful, and I think that's interesting. I think what would be a great paper would be to apply these lessons in a way that achieves it with cheaper compute resources so as to not significantly add to the reservoir update steps and to do so in a truly local way.
>
> In any case, this paper can still be relevant as a step in achieving this goal, so I don't think it's dismissable.

---

> ### Author Response · Authors · 2024-11-27
> **Second Response to Reviewer Feedback**
>
> We sincerely appreciate the positive feedback on the revisions made and are grateful for your adjusted review score. Your insights have undoubtedly enriched the clarity and depth of our paper.
>
> Regarding your comments, we concur that developing local plasticity rules capable of achieving similar decorrelation effects without additional computational overhead is an exciting direction for future research. To clarify, our current approach can be viewed as employing a structural plasticity algorithm that initializes with a blank matrix. This initial condition means that neurons lack local information about other neurons, as they do not receive any inputs initially.
>
> We want to stress that this is still grounded in biological motivation. Our algorithm draws inspiration from activity-dependent structural plasticity, where dendrites and axons grow and retract based on the level of neuronal activation (See references  [1], [2] and [3]). A previous research exploring the computational implications of such structural changes includes studies [4] from which the current algorithm is derived. A crucial element of this research is that integrating this type of unsupervised mechanism with minimal supervised elements (reservoir computing), can lead to significant improvements in data processing efficiency and effectiveness.
>
> Regrettably, due to the word limit constraints of the ICLR format, we initially omitted a detailed discussion and pertinent references on these biological motivations. We now recognize that their inclusion, particularly reference [4], would substantively strengthen the manuscript. We are going to introduce [4] in the next version of the manuscript. Should you see value in expanding upon these biological motivations as well, we are prepared to incorporate a detailed discussion in the revised manuscript.
>
> As we continue to refine our manuscript, we are also developing additional visualizations to more clearly depict the dynamics of our algorithm and its impact on network architecture. Should you have further suggestions that could enhance our paper’s strength or impact, we welcome them eagerly, especially during this extended rebuttal period.
>
> [1] M., & Tetzlaff, C. (2016). Opposing Effects of Neuronal Activity on Structural Plasticity. Frontiers in Neuroanatomy, 10, 75. doi:10.3389/fnana.2016.00075
>
> [2]  Christopher S. Cohan, Stanley B. Kater , Suppression of Neurite Elongation and Growth Cone Motility by Electrical Activity.Science232,1638-1640(1986).DOI:10.1126/science.3715470
>
> [3] Signaling Mechanisms Underlying Reversible, Activity-Dependent Dendrite Formation, Vaillant, Andrew R. et al., Neuron, Volume 34, Issue 6, 985 - 998
>
> [4] T. Cazalets and J. Dambre, "An homeostatic activity-dependent structural plasticity algorithm for richer input combination," 2023 International Joint Conference on Neural Networks (IJCNN), Gold Coast, Australia, 2023, pp. 1-8, doi: 10.1109/IJCNN54540.2023.10191230.

---

> ### Author Response · Authors · 2024-11-29
> **Third Response to Reviewer**
>
> We have incorporated several additional improvements into the final revision:
>
> 1.	Enhanced Motivation: To clearly illustrate the decorrelation mechanism of our approach, we have added an additional subfigure to Figure 2.
>
> 2.	Resulting Architecture Visualization: We have included a detailed visualization of the final network architecture produced by the algorithm. This provides new insights into the clustering and connectivity dynamics shaped by the growth and pruning mechanism. Notably, this analysis allowed us to diagnose the reasons for the variance-HAG model’s underperformance on the JapaneseVowels dataset.
>
> 3.	Extended Contextualization: We have further expanded the biological motivation for this work and, as suggested by another reviewer, extended the contextualization of our contributions within the broader research landscape.
>
> We believe these updates address some of the remaining points raised and enhance the manuscript’s clarity and depth. We wanted to express our gratitude once more for your detailed feedback and for the opportunity to further refine our work and we hope the revisions effectively reflect your comments and insights.

---

### Official Review · Reviewer_MaFX · 2024-11-03

**Soundness:** 2
**Presentation:** 2
**Contribution:** 2
**Rating:** 5
**Confidence:** 4

**Summary:**

In this paper, the authors present Hebbian Architecture Generation (HAG), an innovative method that dynamically adjusts synaptic weights within RNN to enhance the quality of their representations. Experimental results demonstrate that HAG achieves superior performance compared to traditional Echo State Networks across a range of predictive modeling and pattern recognition benchmarks.

**Strengths:**

***1.*** The authors propose a new approach to Echo State Networks called HAG.

***2.*** HAG achieves improved performance compared to the original ESN method.

***3.*** The authors assessed the correlation among neural states using Pearson correlation.

**Weaknesses:**

***1.*** The motivation for the method in the article is unclear; in other words, it is not evident why HAG was proposed, and the article does not provide a clear explanation. Additionally, it does not specify the main problem that the article aims to focus or address.

***2.*** There is no comparison with other methods in the results, only a comparison with the original ESN. This means that the method merely offers an improvement but does not demonstrate how effective it truly is. Additionally, there are no meaningful characteristics, aside from higher accuracy, to differentiate it from other methods.

***3.*** The descriptions of the figures are unclear, making it difficult to understand the significance of the data presented.

**Questions:**

***1.*** The authors need to explain the motivation behind proposing this method, why it was developed, and its significance.

***2.*** The authors should expand their experiments to demonstrate the true effectiveness of their method, as the current experiments only serve as a basic validation. This includes providing more detailed results, explanations of Pearson correlation, and other relevant metrics to substantiate the method's impact.

***3.*** The purpose of Figures 3 and 4 is unclear; a more detailed explanation is needed to clarify the meaning of each subplot.

***4.*** Figure 2 does not provide any more useful information, the different curves are mixed together, and the problem illustrated by this figure is not clear.

---

> ### Author Response · Authors · 2024-11-22
> **Response to Reviewer Feedback**
>
> Thank you for your thoughtful review. Below, we address the key concerns raised and outline the revisions made in response to your feedback:
>
> 1. Motivation of the method: A new section titled “Motivation” has been added to clarify the rationale behind the proposed algorithm. This section elaborates on the main problem the paper aims to address and why HAG was developed as a solution. The discussion now includes references to Cover’s theorem to contextualize the method’s design and significance. This addresses concerns about the unclear motivation for the work.
>
> 2. To strengthen the evaluation, we have added comparisons with IP-ESN : https://linkinghub.elsevier.com/retrieve/pii/S0925231208000519
>
> 3. We added significant improvement to the figures :
> * Figure 2: Figure 2 has been updated to better align with the section’s content. It now provides a clearer visualization of the role of \beta and \rho
> * Figure 3: Adjustments include consistent axis labels across all subplots and an updated caption to clarify the purpose of each subplot.
> * Figure 4:  Similarly, the axis labels and timescales have been standardized across subplots, and the caption has been revised for better clarity.
>
> Questions:
>
> 1. Motivation of the method: As previously said a new section titled “Motivation” has been added to clarify the rationale behind the proposed algorithm. This section elaborates on the main problem the paper aims to address and why HAG was developed as a solution.
>
> 2. More detailed results and explanations : The experimental section has been expanded to include more detailed results, incorporating additional metrics to substantiate the method’s impact.  The details on results is available in Appendix C and D. We also included a clear explanation of Pearson correlation calculations, which is provided in the Appendix A for completeness and transparency.
>
> 3. We added a more detailed caption to each of those figures.
>
> 4. Figure 2 has been updated to illustrate better the dynamics of the algorithm.
>
> We hope this update address the concerns raised and strengthen the paper. Thank you again for your constructive feedback.

---

### Official Review · Reviewer_TZdK · 2024-11-04

**Soundness:** 2
**Presentation:** 3
**Contribution:** 2
**Rating:** 5
**Confidence:** 3

**Summary:**

The paper introduces a Hebbian-inspired algorithm for adaptive reservoir computing networks. The algorithm adjusts synaptic weights in the reservoir network as a function of the correlation between neuron pairs, and their corresponding target activity. The paper reports results that outperform traditional and excitatory-only Echo State Networks.

**Strengths:**

**An interesting Bio-Plausible Approach: Homeostatic Plasticity for Reservoirs**

The main strength of this work lies in its use of homeostasis—a phenomenon prevalent in neural circuits—to improve reservoir learning. The results presented are both promising and impressive, especially given the simplicity of the homeostatic constraint (i.e., maintaining Δz between -1 and 1). I also appreciate the paper’s thorough evaluation across both classification and regression benchmarks.

**Weaknesses:**

**Concerns with Methodology**

- The calculation of Pearson correlation between neuron pairs is unclear.
- The choice of correlation bounds \(-1\) and \(1\) seems arbitrary—can you explain the reasoning behind this?
- What threshold is used to determine "highly correlated neuron pairs" for establishing new connections?
- The statement, "Weight needs to be decreased by \(\delta_w\)," lacks detail—how is \(\delta_w\) calculated?
- Why are connections pruned at random?
- Figure 2 appears arbitrary; it is neither referenced in the text nor adequately explained in its caption.
- The section on Cover's theorem seems irrelevant and unnecessary, as the theorem is not mentioned again after this section.
- How is the target activity \(\rho\) determined?

The lack of a clear methodology undermines confidence in the results. Including pseudocode could greatly enhance transparency. I would consider adjusting my score if the authors address these concerns.

**Questions:**

Could the authors clarify the concept of 'Suitability Characterization'? This is mentioned as a key contribution of HAG.

---

> ### Author Response · Authors · 2024-11-22
> **Response to Reviewer Feedback**
>
> Thank you for your careful reading and detailed review. Below, we address the key concerns raised:
>
> 1.	Pearson Correlation : The text has been expanded to include a detailed explanation of the calculation of Pearson correlation. This additional detail is provided in the Appendix  A for completeness and transparency.
>
> 2.	Choice of Correlation Bounds (-1 and 1): The explanation around the choice of correlation bounds has been clarified. By scaling the deviation (s - \rho) by \beta, \Delta z becomes a normalized measure of deviation. The bounds -1 and 1 provide a symmetric range around zero, which simplifies the decision logic within the algorithm. However, these exact values are not critical, as the scaling by \beta ensures normalization. This revised explanation has been included in the manuscript for better clarity. Figure 2 as also been updated to illustrate this point better.
>
> 3.	“Highly Correlated Neuron Pairs”: we acknowledge the mistakes in explanation by using the term “highly correlated neuron pairs.” The algorithm actually identifies “the most correlated neuron pairs” (those with the highest Pearson correlation), and from these, selects a neuron to create a connection. This has been corrected and clarified in the manuscript.
>
> 4.	Clarification of Parameters (\delta_w, \rho, \beta): We acknowledge that the original text may have caused confusion regarding the roles of these parameters. These values are, in fact, hyper-parameters, as highlighted in Table 5 ( previously Table 1 but now moved in the appendix C.), where we now explicitly associate each variable name with its corresponding notation. This should help clarify their purpose and significance. A detail account of hyper parameter optimization results can be found in newly added appendix C.
>
> 5.	Random Pruning of Connections: the motivation behind HAG is to improve upon the limitations of random reservoirs by creating a structured reservoir that maximizes the benefits of high-dimensional projection by increasing the effective dimensionality and reducing redundancies in the reservoir’s representation. This is done by incrementally connecting nodes that are highly correlated to create new dynamics (the dynamics being changed by the added connection). However when node arrives to saturation, we want to lower a connection. Determining which specific connection contributes least to the neuron’s current dynamics is challenging and computationally expensive, as it would require analyzing the influence of each connection on the neuron’s activity in the context of the entire network. Instead of undertaking this complex task, we employ a random pruning strategy. By randomly selecting one of the neuron’s connections to weaken or remove, we simplify the pruning process while avoiding potential biases that could arise from systematic pruning methods.
>
> 6.	Updates to Figure 2: Figure 2 has been updated to better align with the content of the corresponding section. It now visualizes the roles of \beta and \rho, making it more informative and directly relevant to the discussion.
>
> 7.	Cover’s Theorem:  We agree that the sections on Cover’s theorem were not effectively integrated into the main discussion. To address this, we have condensed this content and included a brief mention of Cover’s theorem in a newly added “Motivations” section. This section provides context for the algorithm without detracting from the main focus of the paper.
>
> 8. Other points :
> * Suitability Characterization: The concept of “Suitability Characterization” refers to the ability to assess and quantify how well a reservoir is suited for a particular task based on measurable network properties, rather than relying on the tautological notion that a reservoir is suitable if it simply yields accurate models. This was a point of discussion in Jaeger’s 2005 paper “Reservoir Riddle,” where he pointed out the need for concrete criteria to evaluate reservoir suitability.
> * Pseudo code: Overall the description of the algorithm have been improved as we have rewritten parts of the text and added a step-by-step listing of the HAG algorithm in pseudocode format, which is now included in Appendix B.
>
> We believe these updates address the key concerns and significantly improve the clarity and transparency of the methodology. Thank you again for your constructive feedback, which has been invaluable in enhancing the quality of the manuscript.

---

> > ### Comment · Reviewer_TZdK · 2024-11-27
> >
> > I have acknowledged the authors' work to address the manuscript's comments. I will slightly adjust my score since many of my concerns have been addressed.
> >
> > While I have made adjustments to my evaluation, I still find it challenging to determine where this work fits within the broader reservoir computing literature. The manuscript's motivations, as they pertain to existing research, are not clearly articulated, leaving its contribution ambiguous. Consequently, although I am revising my score, I am unable to recommend acceptance of this manuscript at this time. That said, I will also adjust my confidence level accordingly to reflect this reassessment.

---

> > > ### Author Response · Authors · 2024-11-27
> > > **Second Response to Reviewer Feedback**
> > >
> > > Thank you for re-evaluating our manuscript and adjusting your score. We greatly appreciate your constructive feedback and continued engagement.
> > >
> > > 1. To better clarify the positioning of our approach within the reservoir computing literature, we emphasize its novelty in leveraging correlation-based synaptic adjustments within a homeostatic framework. While our work references related studies such as [A], [B], and [C], we acknowledge the need to expand this discussion. Specifically, we will incorporate findings from [D], which explores structural plasticity but demonstrates limited results compared to our approach. We also recognize the need for clearer contextualization in our writing and will revise the manuscript accordingly to highlight the significance of our contributions.
> > >
> > > 2. To strengthen the manuscript’s motivation, we propose the following enhancements:
> > > * Regarding the motivations of this research, as noted in our response to another reviewer, our integration of unsupervised, activity-dependent structural plasticity mechanisms (as discussed in [1], [2], and [3]) with minimal supervised elements builds on the concept that such biologically plausible adaptations can significantly enhance the efficiency and effectiveness of reservoir computing systems. This approach aligns with the hypothesis in [4], which suggests that biologically inspired adaptations can achieve remarkable efficiency and performance with significantly less supervision compared to conventional neural network models.
> > > * We will further elaborate on the role of decorrelation as a method for increasing linear separability, grounding our approach in well-established mathematical principles.
> > >
> > > If you have additional suggestions or identify other directions in which we could further extend this work, we would welcome your input during the extended rebuttal period.
> > >  Thank you once again for your valuable feedback.
> > >
> > > **References :**
> > >
> > > [A] Guillermo B. Morales, Claudio R. Mirasso, and Miguel C. Soriano. Unveiling the role of plasticity rules in reservoir computing. Neurocomputing, 2021. doi: 10.1016/j.neucom.2020.05.127
> > >
> > > [B] SORN: A self-organizing recurrent neural network, October 2009, Frontiers in Computational Neuroscience 3:23, DOI:10.3389/neuro.10.023.2009
> > >
> > > [C] H. Jaeger. Reservoir riddles: suggestions for echo state network research (extended abstract). In Proceedings. 2005 IEEE International Joint Conference on Neural Networks, 2005., IJCNN-05. IEEE, 2005. doi: 10.1109/ijcnn.2005.1556090.
> > >
> > > [D] T. Cazalets and J. Dambre, "An homeostatic activity-dependent structural plasticity algorithm for richer input combination," 2023 International Joint Conference on Neural Networks (IJCNN), Gold Coast, Australia, 2023, pp. 1-8, doi: 10.1109/IJCNN54540.2023.10191230.
> > >  [1] M., & Tetzlaff, C. (2016). Opposing Effects of Neuronal Activity on Structural Plasticity. Frontiers in Neuroanatomy, 10, 75. doi:10.3389/fnana.2016.00075
> > >
> > > [2]  Christopher S. Cohan, Stanley B. Kater , Suppression of Neurite Elongation and Growth Cone Motility by Electrical Activity.Science232,1638-1640(1986).DOI:10.1126/science.3715470
> > >
> > > [3] Signaling Mechanisms Underlying Reversible, Activity-Dependent Dendrite Formation, Vaillant, Andrew R. et al., Neuron, Volume 34, Issue 6, 985 - 998
> > >
> > > [4] Zador, A.M. A critique of pure learning and what artificial neural networks can learn from animal brains. Nat Commun 10, 3770 (2019). https://doi.org/10.1038/s41467-019-11786-6

---

> > > > ### Author Response · Authors · 2024-11-29
> > > > **Third response to reviewer**
> > > >
> > > > We have implemented several additional improvements to address the points you raised and to further clarify the unique contributions of our work:
> > > >
> > > > 1.	**Enhanced Motivation and Visualization of Decorrelation Mechanisms**
> > > > To better elucidate the decorrelation mechanism central to our approach, we have added an additional subfigure to Figure 2. This new subfigure illustrates how the algorithm leverages Pearson correlation to guide synaptic adjustments, creating a more direct connection between the theoretical framework and its practical implementation.
> > > >
> > > > 2.	**Broader Contextualization of Contributions**
> > > > We have significantly expanded the discussion on how our approach fits within the reservoir computing literature, incorporating several new references to better position this work within the field. Furthermore, we have strengthened the biological motivation, providing a more robust scientific foundation for our methodology.
> > > >
> > > > We hope these final additions further strengthen the manuscript and clarify the novelty and significance of our contributions to the field.

---

### Official Review · Reviewer_1oGH · 2024-11-04

**Soundness:** 3
**Presentation:** 2
**Contribution:** 2
**Rating:** 5
**Confidence:** 4

**Summary:**

The paper presents Hebbian Architecture Generation (HAG), a method for dynamically constructing reservoir computing networks. Unlike standard Echo State Networks (ESNs) with static random connectivity, HAG dynamically forms connections between neurons that have high Pearson correlation. Two variants are introduced: mean-HAG and variance-HAG, which adjust weights based on mean activity and standard error respectively. Starting from a blank connectivity matrix, the method builds excitatory connections based on Pearson correlations. Testing on classification and prediction tasks shows improvements over ESNs in classification.

**Strengths:**

- The use of Pearson correlation to form connections is novel to my knowledge
- The empirical results on the tested datasets seem reasonable

**Weaknesses:**

- The method is not explained very clearly. The exact training algorithm would have been clearer with an explicit algorithm listing.
- The empirical evidence seems to indicate the method works ok, but comparisons with NG-RC [1] would have been useful.
- The sections on Cover's theorem adds very little to the overall paper, and could have been explained in a sentence.
- The implications of analysis in Section 5.1 is not clear

[1] https://www.nature.com/articles/s41467-021-25801-2

**Questions:**

- Hows the mean and std calculated? Using what data?
- The Intrinsic plasticity mechanism described in section 3.1.1 also sounds like a homeostasis mechanism.
- E-ESN is not explained in the results tables

Minor:
- Pg. 132-139: There's a "First" and "Third" but no "Second".

---

> ### Author Response · Authors · 2024-11-22
> **Response to Reviewer Feedback**
>
> Thank you for your careful reading and thoughtful review. Below, we describe how the main points raised have been addressed and we answer the questions.
>
> Answers to Weaknesses :
>
> 1.	Clarity of Method Explanation: We acknowledge that the method’s explanation was not as clear as it should have been. In response, we have rewritten parts of the text and added a step-by-step listing of the HAG algorithm in pseudocode format, which is now included in Appendix B.
>
> 2.	Comparison with NG-RC: Thank you for suggesting a comparison with NG-RC [1]. While NG-RC is an important framework, it differs from our work in that it does not utilize a reservoir per say, whereas our focus is on proposing an initialization method for reservoir-based systems. Additionally, the resource requirements for NG-RC, particularly in memory, are significantly different from those of our method. However, we appreciate the value of such a comparison and have now included an experimental comparisons with IP-Reservoir [2], which is more closely related to our approach.
>
> [1] https://www.nature.com/articles/s41467-021-25801-2
>
> [2] https://linkinghub.elsevier.com/retrieve/pii/S0925231208000519
>
> 3. As noted by several reviewers the link between the Cover's theorem and the rest of the work was not properly done. Following your advice I removed it and only mention it in a newly added section about the « Motivations » behind the algorithm that were also asked by some reviewers.
>
> 4.	Implications of Section 5.1 Analysis:  Our goal in Section 5.1 was to illustrate how the Hebbian Architecture Generation (HAG) algorithm enhances the reservoir’s dynamics by reducing the correlation among neuron activations over time. In Section 5.1, we analyze the Pearson correlation among neuron activation states within the reservoir. Initially, the neurons exhibit high correlation, meaning they respond similarly to inputs. However, as the HAG algorithm dynamically adjusts the network’s connectivity based on neuron correlations, we observe a decline in the mean Pearson correlation.
> This decline in correlation implies that neurons are becoming less synchronized and are diversifying their responses to the input signals. This diversification effectively increases the dimensionality of the reservoir’s state space. According to Cover’s theorem, projecting input data into a higher-dimensional space enhances the likelihood that complex patterns become linearly separable.
> By reducing redundancy and promoting diverse neuron activations, the HAG algorithm enriches the reservoir’s dynamic representation of input data. This leads to improved pattern recognition and predictive performance, as evidenced by our experimental results.
> We also includes an additional section in the appendix where we explain in more details how Fig 5. was formed.
>
> Answers to Questions:
>
> * Mean and Standard Deviation Calculations: The mean and standard deviation are calculated for each neuron over the period T. Notations have been updated to make this clearer.
>
> * Intrinsic Plasticity Mechanism: Intrinsic plasticity is indeed a homeostatic mechanism. We clarified this point by emphasizing that variance-HAG and this homeostatic mechanism target different moments of the data (variance for variance-HAG, and the mean for the homeostatic). We change the name from Intrinsic plasticity to homeostatic plasticity mechanism to avoid confusion with the IP-ESN [2] that we added for comparison with our algorithm.
>
> * E-ESN: We recognize that the explanation of E-ESN in Section 4 was too brief. It has been expanded to ensure clarity, specifying that it is an ESN with only positive weights, to ensure fair comparison with HAG-generated networks that also have only positive weights.
>
> Minor Issues:
>
> * The inconsistency in “First” and “Third” without a “Second” (pages 132-139) has been corrected for logical flow.
>
> We hope these changes address the concerns raised and strengthen the manuscript. Thank you for your constructive feedback.

---

### Comment · Area_Chair_9ZWp · 2024-11-25

Dear reviewers,

A reminder that **November, 26** is the last day to interact with the authors, before the private discussion with the area chairs. At the very least, please acknowledge having read the rebuttal (if present). If the rebuttal was satisfying, please improve your score accordingly. Finally, if you have concerns that might be solved in time, this is the last chance before moving on to the next phase.

Thanks,
The AC

---

### Note · Authors · 2025-04-04

I have read and agree with the venue's withdrawal policy on behalf of myself and my co-authors.

---

### Meta-Review · Area_Chair_9ZWp · 2024-12-13

**Metareview:**

The paper introduces a mechanism to adapt the reservoir in an echo state network based on the Pearson correlation coefficients between the units, which they term as "Hebbian Architecture Generation".

All reviewers were highly critical of the work. After the rebuttal phase, the scores include three marginal rejects and one marginal accept. Most reviewers were concerned about the quality of the presentation and the fact that many key parts of the paper were unclear. In particular, one reviewer very convincingly argued that "Hebbian" is a misnomer here, and that the method was not properly characterized.

Finding rules for increasing the plasticity of reservoirs is an interesting research direction. However, it is clear that the paper has significant issues that should be resolved, and the reviewers have provided multiple directions for improving the work.

**Additional Comments On Reviewer Discussion:**

- **Reviewer tQ4Q** made a very thorough and complete review, highlighting in particular (a) unclear motivation and descriptions in the paper, and (b) misuse of the term "Hebbian", as the method does not involve product of activations and it is not local in space or time (as it requires a global statistic of the time series). After the rebuttal discussion, point (a) was partially solved but point (b) remained open. *I fully agree with this evaluation* and it weighted a lot in my final decision.

- **Reviewer MaFX** made a short review highlighting a lack of motivation and a lack of comparisons. They haven't interacted in the rebuttal. *This review had little influence on my final score* as I found it hasty and lacking details.

- **Reviewer TZdK** had many issues related to the unclear methodology. While some of these were clarified in the rebuttal, there is a feeling that the contributions are ambiguous and that "it [is] challenging to determine where this work fits within the broader reservoir computing literature". This review complements that of reviewer tQ4Q.

- **Reviewer 1oGH** was another hasty review, highlighting concerns on the presentation, on a lack of comparison, and on some unclear points. They did not interact in the rebuttal phase. Similarly to reviewer MaFX, *this review had little influence on my final score*.

---

### Decision · Program_Chairs · 2025-01-22

Reject